# Applications of Artificial Intelligence in Acute Promyelocytic Leukemia: An Avenue of Opportunities? A Systematic Review

**DOI:** 10.3390/jcm14051670

**Published:** 2025-03-01

**Authors:** Mihnea-Alexandru Găman, Monica Dugăeşescu, Dragoş Claudiu Popescu

**Affiliations:** 1Faculty of Medicine, “Carol Davila” University of Medicine and Pharmacy, 050474 Bucharest, Romania; monica.dugaesescu@drd.umfcd.ro (M.D.); dragos-claudiu.popescu@drd.umfcd.ro (D.C.P.); 2Department of Hematology, Centre of Hematology and Bone Marrow Transplantation, Fundeni Clinical Institute, 022328 Bucharest, Romania; 3Department of Cellular and Molecular Pathology, Stefan S. Nicolau Institute of Virology, Romanian Academy, 030304 Bucharest, Romania; 4Clinical Laboratory Department, Fundeni Clinical Institute, 022328 Bucharest, Romania; 5Department of Hematology, Emergency University Hospital of Bucharest, 050098 Bucharest, Romania

**Keywords:** acute promyelocytic leukemia, artificial intelligence, machine learning, deep learning, acute leukemia, acute myeloid leukemia, cancer, clinical decision support system, convolutional neural networks, computer vision, digital health, large language models, clinical oncology

## Abstract

***Background***. Acute promyelocytic leukemia (APL) is a subtype of acute myeloid leukemia defined by the presence of a genetic abnormality, namely the *PML::RARA* gene fusion, as the result of a reciprocal balanced translocation between chromosome 17 and chromosome 15. APL is a veritable emergency in hematology due to the risk of early death and coagulopathy if left untreated; thus, a rapid diagnosis is needed in this hematological malignancy. Needless to say, cytogenetic and molecular biology techniques, i.e., fluorescent in situ hybridization (FISH) and polymerase chain reaction (PCR), are essential in the diagnosis and management of patients diagnosed with APL. In recent years, the use of artificial intelligence (AI) and its brances, machine learning (ML), and deep learning (DL) in the field of medicine, including hematology, has brought to light new avenues for research in the fields of blood cancers. However, to our knowledge, there is no comprehensive evaluation of the potential applications of AI, ML, and DL in APL. Thus, the aim of the current publication was to evaluate the prospective uses of these novel technologies in APL. ***Methods***. We conducted a comprehensive literature search in PubMed/MEDLINE, SCOPUS, and Web of Science and identified 20 manuscripts eligible for the qualitative analysis. ***Results***. The included publications highlight the potential applications of ML, DL, and other AI branches in the diagnosis, evaluation, and management of APL. The examined AI models were based on the use of routine biological parameters, cytomorphology, flow-cytometry and/or OMICS, and demonstrated excellent performance metrics: sensitivity, specificity, accuracy, AUROC, and others. ***Conclusions***. AI can emerge as a relevant tool in the evaluation of APL cases and potentially contribute to more rapid screening and identification of this hematological emergency.

## 1. Introduction

The 5th edition of the World Health Organization (WHO) Classification of Hematolymphoid Tumors recognizes acute promyelocytic leukemia (APL) as a subtype of acute myeloid leukemia (AML) defined by the presence of a genetic abnormality, namely the *PML::RARA* gene fusion [1]. The pathogenesis of APL is driven by the occurrence of a reciprocal and balanced translocation between chromosome 17 (the situs of the retinoic acid receptor α gene) and chromosome 15 (where the promyelocytic leukemia gene resides) which results in the production of the *PML::RARA* oncogenic fusion protein [2,3]. Needless to say, cytogenetic and molecular biology techniques, i.e., fluorescent in situ hybridization (FISH) and polymerase chain reaction (PCR), are essential in the diagnosis and management of patients diagnosed with APL. Amongst blood cancers, APL remains a veritable emergency in the hematology clinic due to the risk of early death and coagulopathy if unrecognized and/or left untreated [2,3,4,5,6,7]. Nevertheless, conventional morphology, which can detect the presence of atypical promyelocytes with numerous, bright azurophilic granules, and intracellular inclusions organized in faggots or bundles (Auer rods) on the peripheral blood and bone marrow aspirate smears, as well as multiparametric flow-cytometry, which highlights high side scatter, positivity for CD13, CD33, CD117, and MPO, and negativity for CD34 and HLA-DR, remain essential tools in APL diagnosis, with cytomorphology being the cheapest and widely available laboratory technique [2,6,8,9,10].

Recent years have brought undeniable progress in the diagnosis and management of hematological malignancies. Moreover, the application of artificial intelligence (AI) and its branches, such as machine learning (ML) or deep learning (DL), in the field of medicine, including hematology, has brought to light new avenues for research in the fields of leukemia, lymphoma, myeloma, myeloproliferative (MPN), or myelodysplastic neoplasms (MDS), to name a few [11]. There seems to be a growing interest in AI in the evaluation of hematologic malignancies as depicted in Figure 1.

Examples of AI applications in hematology include, but are not limited to, the use of systems designed to analyze images in cytomorphology and pathology, of bioinformatics to refine the results of genomics and other omics-based approaches, of language processing systems, utilized to rapidly assess real-world data such as electronic health records, of decision support systems meant to propose diagnostic and treatment algorithms, as well as of medical device-related systems, e.g., smartwatches with sensors capable of detecting changes in body parameters and signal potential treatment-related complications [13]. ML has proven excellent performance metrics (sensitivity, specificity, and accuracy) in the diagnosis, classification, and prognostication of MDS and AML using a diverse range of information obtained from clinical examinations as well as laboratory data, i.e., cytomorphology (peripheral blood and bone marrow aspirate smears), cytogenetics, RNA sequencing and/or genomics [13,14]. The assessment of peripheral and blood marrow smear images using AI, and, in particular, convolutional neural networks (CNNs), demonstrates over 90% accuracy in diagnosing AML and MDS, as well as distinguishing between these two myeloid malignancies and chronic myeloid leukemia (CML), chronic lymphocytic leukemia (CLL), or acute lymphoblastic leukemia (ALL) [13]. Moreover, AI has proven useful in estimating prognosis in MDS and AML. Researchers have made an appeal to gradient-boosting to establish decision tree machine learning libraries that could predict MDS development one year before the actual diagnosis, the transformation of MDS to AML, overall survival or complete remission, or inform about AML prognosis solely based on *FLT3-ITD* gene mutation status [14]. In addition, AI can be used to predict treatment response, with CNNs, decision trees, operators, and classifiers being employed to predict response to hypomethylating agents, allogeneic bone marrow transplantation, or the risk of AML relapse [14]. A comprehensive review of the applications of AI in MDS highlighted that this innovative technique can be used to assess complete blood counts, peripheral blood and bone marrow aspirate smears, as well as flow-cytometry data, and establish a diagnosis of MDS, differentiate MDS from aplastic anemia or AML, and detect dysplastic cells of all lineages as well as myeloblasts. This is of particular importance as the differential diagnosis between hypoplastic MDS and aplastic anemia is often cumbersome and both diseases require different treatment strategies. Additionally, establishing myeloblast thresholds is extremely relevant in distinguishing MDS from AML and in selecting the most appropriate therapy to prevent leukemic transformation [14]. CNNs were the most frequent AI models used in the evaluation of MDS, were internally validated, and displayed excellent performance metrics and lower percentages of misclassifications. However, limitations of AI in MDS included a lack of external validation and standardized methodology for the use of flow-cytometry in MDS, small sample sizes, as well as the need for supervision [14]. Furthermore, AI has proven significant potential in AML management, as it is able to discriminate between hematopoietic stem cells and leukemia stem cells based on cytomorphology and flow-cytometry reports with an accuracy of nearly 93.5%. These findings further solidify the hypothesis that AI can become practical in the prediction of response to AML treatment, including early relapse post-allogeneic hematopoietic stem cell transplantation [15]. Complications can also be predicted using AI algorithms in AML. Doknic and collaborators developed an ML-based method to assess the risk of venous thrombosis in AML, revealing that male sex, history of thrombotic events, international normalized ratio, hematopoietic cell transplantation-specific comorbidity index, and intensive chemotherapy are risk factors for the onset of thrombotic complications in this myeloid malignancy [16]. In children, AI has been employed to quantify the risk of viral and bacterial infections in AML and ALL with an accuracy of nearly 80%. Using a framework based on ML (decision trees combined with SemNet 2.0), Al-Hussaini and his team have demonstrated that there are several factors that can influence the risk of infectious complications in pediatric patients with ALL or AML: race, type of acute leukemia, association with Down syndrome or lupus, central nervous system involvement at diagnosis, National Cancer Institute risk group, chemotherapy regimen, and course, glucose, zinc or iron metabolism, and growth factor levels [17].

Similarly, AI seems promising in the evaluation of MPN as well. In *BCR::ABL1*-negative MPN, ML and CNN have displayed extremely good performance metrics in diagnosing MPN, differentiating between MPN subtypes, and, in particular, distinguishing essential thrombocythemia from prefibrotic primary myelofibrosis, as well as in predicting prognosis, occurrence of thrombotic events or treatment response to hydroxyurea. These AI tools appeal to clinical data, routine laboratory parameters, peripheral blood smears, bone marrow biopsy samples, transcriptomics, and genomics, and led to rapid assessment of cases and accurate discrimination between MPN subtypes. However, limitations to the AI instruments in MPN include the need for larger patient cohorts for training and validation, as well as the requirement for human supervision for many of these tools [18]. More recent investigations have delineated that AI can accurately predict the development of thrombosis in subjects with polycythemia vera who were prescribed treatment with hydroxyurea. In the aforementioned AI algorithm, the onset of thrombotic complications could be predicted by using only three complete blood count-derived variables: red cell distribution width, neutrophil percentage, and lymphocyte percentage [19]. In CML, ML instruments have been applied to investigate the interactions established between the BCR-ABL1 protein and tyrosine kinase inhibitors in the scope of discovering novel drugs and therapeutic targets [20].

Furthermore, the use of AI platforms has also been warranted in the assessment of chronic lymphoproliferative disorders. In multiple myeloma (MM), AI has been employed to establish an early diagnosis, prognosis, disease staging, identification of bone lesions and molecular aberrations, as well as to guide therapy selection [21]. For example, CNNs and decisional trees have been combined with data obtained from routine hematological and biochemical blood tests, flow-cytometry, and spectroscopy to accurately establish an early diagnosis in MM, as well as to stage the disease [21]. Moreover, information collected from imaging (PET-CT, CT, or MRI) has been connected with AI techniques (CNNs, random forest, support vector machines) to quantify the level of skeletal involvement in MM, as well as to discriminate between bone lesions and vertebral metastases of other causes [21]. Response to treatment has also been estimated with the aid of AI. Random forests, support vector machines, simulations of treatment learning signatures, and multilearning treatment approaches have been combined with clinical and genomic data to estimate response to proteasome inhibitors, immunomodulatory agents, corticosteroids, and chemotherapy [21]. ML models have also been developed to predict the risk of infectious complications which account for a significant proportion of deaths in MM. Aided by the use of random forest, decision tree, and gradient boosting algorithms, Mikulski and team discovered that low platelet distribution width values predict the occurrence of pneumonia in patients with MM who had received regimens containing the anti-CD38 monoclonal antibody daratumumab [22]. Concerning lymphoma, a recent meta-analysis of diagnostic studies has highlighted that AI algorithms displayed a specificity and sensitivity of 94% and 87%, respectively, in the detection of lymphoma based on imaging data [23]. PET-CT findings have also been combined with ML models to decide whether there is a degree of bone marrow involvement in Non-Hodgkin’s lymphoma, a piece of information that is crucial for disease staging and selection of the most appropriate therapy [24]. Moreover, proteomics has been connected with bioinformatics to quantify the risk of progression of follicular lymphoma. Using unsupervised ML models, Hemmingsen et al. have revealed that patients with low expression of STING1 and IDH2 are at higher odds of progressing to diffuse large B-cell lymphoma (DLBCL) [25]. In addition, AI seems to be superior to the reports given by expert pathologists in predicting the transformation of CLL or follicular lymphoma to DLBCL [26]. ML and natural language processing have also been employed to analyze the electronic health records of over 500 CLL patients from Spain in order to produce real-world evidence of the clinical characteristics and therapy patterns of this country in an attempt to solidify the argument that there is a need for personalized treatment in hematological malignancies [27]. Another field of hematology where AI has been investigated is hematopoietic stem cell transplantation (HSCT). Random forest and decision tree-based approaches have successfully appreciated the risk of relapse and the prognosis of patients who had undergone this procedure, and it has been hypothesized that ML will be able in the near future to assist hematologists in selecting the optimal conditioning regimen in HSCT [28]. For example, the relapse rate at 2 years following allogeneic HSCT for AML or ALL in children has been estimated using ML. The random forest approach exhibited an accuracy of 85% for ALL and 81% for AML, a sensitivity of 85% for ALL and 75% for AML, as well as a specificity of 89% for ALL and 100% for AML [29]. AI methods have also enhanced the prediction of post-HSCT complications, e.g., the onset of acute kidney injury. Musial and collaborators applied a random forest classifier to a dataset of pediatric subjects who had undergone HSCT and concluded that the estimated glomerular filtration rate pre-HSCT and in the early period following HSCT, administration of methotrexate use, acute graft versus host disease, and the development of viral infections are the most accurate predictors of acute kidney injury in HSCT recipients [30].

While a vast spectrum of AI tools are currently under development and evaluation for medical purposes, in recent years, several AI-based instruments have been implemented in healthcare services. In oncology, AI has been used, for example, in medical imaging, pathology, tumor characterization, patient triage, and survival prediction [31]. For example, McKinney et al. have reported in Nature the development of an AI-based system for breast cancer screening that can surpass medical imaging experts in detecting breast cancer on mammograms. When used for this purpose in a double-reading process, this AI system reduced the workload of the second reader by 88% [32]. In hematology laboratories, AI-assisted tools, such as AI-assisted digital scanners, have been widely and successfully used in clinical workflows. According to Salib et al., with the aid of AI, the diagnosis of hematological disorders has been improved by the implementation of AI-assisted Scopio Labs x100 Scanner, minimizing subjectivity related to cell counting and classification, and saving time due to the pre-classification of morphological elements [33].

Hematological malignancies heavily depend on laboratory techniques that involve the morphological identification of abnormal cells. These initial assessments are often refined through more advanced investigations, such as immunophenotyping. In the case of acute leukemias, a timely approach is critical for diagnosis and treatment. APL, in particular, requires immediate treatment intervention, as the prompt administration of all-trans retinoic acid (ATRA) can prevent potentially life-threatening complications [4]. This urgent need for rapid diagnosis has led researchers to explore the use of AI to identify APL cases more quickly and efficiently. Unlike traditional methods, many studies have focused on leveraging CNNs, which have proven effective in processing medical imaging data. This has made them well suited for analyzing cellular input data from larger patient cohorts [34].

The main advantages of these AI models include their ability to scale by processing high volumes of data, which would otherwise require significant manual review by trained healthcare professionals. Additionally, CNNs offer consistent results and can detect even subtle morphological variations. Perhaps most importantly, AI can enhance accessibility in areas that lack dedicated laboratory facilities for routine hematopathology assessments. Thus, AI can play a crucial role in the screening phase for APL, serving as an efficient tool for analyzing large sample sizes and identifying profiles that may suggest the presence of this hematological emergency [34].

Nevertheless, to our knowledge, no review has explored the potential applications of AI, ML, and DL in APL. Thus, the aim of the current publication was to evaluate the prospective uses of these novel technologies in APL, based on an overview of the current status of the development and implementation of AI tools with applications in the diagnosis and case management for APL. More specifically, our objective was to highlight the available integrable approaches that appeal to conventional APL diagnostic instruments (cytomorphology, flow-cytometry, molecular biology/omics, and routine laboratory parameters) and AI, with a focus on AI branches such as ML, DL, and CNNs and their performance metrics, in the assessment of APL. Therefore, we conducted a comprehensive narrative review of the current literature published between 2016 and 2025 and indexed in three reference scientific databases for medicine (PubMed/MEDLINE, Web of Science, and SCOPUS) that investigates for the first time the applications of AI in APL.

## 2. Materials and Methods

The current review was conducted in agreement with the “Preferred Reporting Items for a Systematic Review and Meta-Analysis” (PRISMA) norms [35]. The research question behind the elaboration of this manuscript was: “Can AI be useful in the assessment of APL?”.

*Information sources and search strategy*. We developed a search strategy in three databases of reference for medicine (PubMed/MEDLINE, SCOPUS, and Web of Science) using the following specific keywords and word combinations: (“acute promyelocytic leukemia” OR “acute promyelocytic leukaemia”) AND (“artificial intelligence” OR “machine learning” OR “deep learning” OR “neural networks” OR “clinical decision support system” OR “computer vision” OR “digital health” OR “large language models” OR “random forest”). The Polygot Search Translator was used to convert the search strategy query between databases [36]. No restrictions were applied to the search strategy in terms of language. In terms of publication timeline, we included manuscripts published from the inception of the aforementioned databases until 10 February 2025.

*Eligibility criteria*. The following inclusion criteria were applied: (1) the included manuscripts were either original research articles or research letters; (2) the study groups of these investigations included confirmed APL cases; (3) the examined studies investigated the use of any AI branch (ML, DL, CNNs, etc.) in APL; (4) the manuscripts were published in a language known to the authors of the current manuscript (English, French, Italian, German, Romanian); (5) the full-texts of the papers could be retrieved. We opted for the following exclusion criteria: (1) the manuscripts were excluded if they were designed as reviews, book chapters, case reports, meeting abstracts, animal studies, or other types of manuscripts; (2) studies conducted on laboratory animals; (3) the diagnosis of APL was not confirmed, e.g., the subjects were diagnosed with non-APL AML or APL-like AML; (4) the papers were published in a language not spoken by the authors; (5) the full-texts of the manuscripts could not be retrieved; (6) the investigations did not report sufficient data on the outcomes of interest (AI models, performance metrics).

*Study selection*, *data collection*, *outcomes of interest*, *risk of bias*, *effect measures, and synthesis methods.* Three researchers independently assessed the titles and abstracts of the selected publications according to the aforementioned search strategy. Quality assessment was conducted using the Quality Assessment of Diagnostic Accuracy Studies version 2 (QUADAS-2) instrument [37]. Data were centralized and analyzed using Microsoft Office Excel 2013 in dedicated spreadsheets. The following information was extracted from the original studies: the type of AI model developed, the input data required to generate predictions/results, the training, testing, and/or validation datasets (data description and number of samples), the reported characteristics of performance for the AI model—in relation to the stage of development and/or testing. The main performance metrics considered were as follows: sensitivity (the rate of true positive responses), specificity (the rate of true negative answers), accuracy (the sum of true positive and true negative responses divided by the number of total observations), and the AUROC (area under the receiver operating characteristic curve). When available, other performance characteristics, e.g., precision, were collected. We used the performance metrics to briefly compare the performance of the AI model clusters investigated by the authors, outlining the subtle differences. If unavailable and data allowed it, we calculated the performance metrics based on the following formulas [38]:
-Sensitivity (True Positive Rate, TPR) = (True Positives)/(True Positives + False Negatives) (it measures the proportion of actual positives that are correctly identified);-Specificity (True Negative Rate, TNR) = (True Negatives)/(True Negatives + False Positives) (the ratio of correctly predicted negative observations to all actual negatives; it measures the proportion of actual negatives that are correctly identified by the model);-Accuracy = (True Positives + True Negatives)/Total Observations (the ratio of correctly predicted observations to the total observations; a common measure of the overall performance of a classification model);-AUROC = a measure of the ability of a classifier to distinguish between classes. It plots the TPR against the False Positive Rate (FPR) at various threshold settings. It is positively correlated with the performance of the analyzed model.

## 3. Results

*Literature search*. In total, 109 potentially eligible manuscripts were detected in the PubMed/MEDLINE, SCOPUS, and Web of Science databases. Of these, 57 papers were excluded as duplicates and the title and/or abstract of the remaining 52 papers were screened. Following this step, an additional 32 papers were excluded and 20 manuscripts were screened to establish whether they met the inclusion criteria. Finally, from the results retrieved through the described search strategy, 20 articles met the inclusion criteria. The articles presented the development and evaluation of AI-based tools with applications in the field of APL. Quality assessment did not reveal a high degree of bias or applicability concerns. The flow diagram of the study and selection procedure is depicted in Figure 2.

### 3.1. Diagnostic Tools Based on Morphological Features

Manescu et al. evaluated a deep learning-based tool, Multiple Instance Learning for Leukocyte Identification (MILLIE), in the diagnosis of acute leukemia. The weakly supervised CNN model automatically classified leukemia cases into subtypes through reading of peripheral blood films and bone marrow aspirates, without individual cell identification training. Being trained with publicly available datasets to differentiate between AML and ALL based on the presence of myeloblasts or lymphoblasts, the tool was further developed to establish APL diagnosis through malignant promyelocyte recognition. Its AUROC for APL detection based on peripheral blood films was 0.94 ± 0.04 and for bone marrow aspirate smears was 0.99 ± 0.01 [39]. The training and testing datasets, the steps of evaluation, and the performance metrics of MILLIE are thoroughly described in Table 1.

Another DL model was developed and evaluated by Eckardt et al. to identify APL cases based on bone marrow aspirate smears. The CNN model was trained and tested with 51 APL, 1048 non-APL AML, and 236 normal bone marrow aspirate smears. APL detection (versus non-APL AML cases) achieved an AUROC of 0.8575. When the comparison was performed against healthy individuals’ bone marrow aspirate smears, the AUROC was 0.9585. Both the mean average precision and the mean average recall were 0.97. From the image upload, a potential diagnosis was obtained in 45.3 s [40].

Ouyang et al. described the development and evaluation of a CNN diagnostic system for APL from bone marrow aspirate smear images. This model was trained and validated (80%:20%) on a retrospective dataset comprising 12,215 images and further tested on another retrospective dataset of 1289 bone marrow aspirate smear films. After several steps of model improvements, an average precision of 62.5% and an average recall of 84.1% were obtained [41]. Qiao et al. developed a three-step pipeline centered on CNN technology to diagnose APL from peripheral blood smears. Firstly, the cell classification model was validated on two five-fold cross-validation models derived from two publicly available datasets comprising 14,910, and 7695 leukocyte images, obtaining an AUROC of 0.9977 ± 0.0003 and a precision of 97.65%, respectively, an AUROC of 0.9914 ± 0.0026, and a precision of 96.53% for APL diagnosis. A clinical dataset (6798 leukocytes) was used to validate the whole pipeline, obtaining a precision of 99.2% [42].

ALNet, another CNN model designed for the diagnosis of acute leukemia using peripheral blood smear images, was trained and tested (85%:15%) with 16.450 images of single cells from 731 blood smears, obtained from 148 acute leukemia patients, 100 healthy individuals, and 191 persons diagnosed with viral infections. ALNet correctly predicted the APL diagnosis for all 14 patients who in fact suffered from this AML subtype [43].

Sidhom et al. also focused on developing a CNN model to diagnose APL based on peripheral blood smear images. The instrument was evaluated using a discovery cohort and an independent prospective validation cohort, which included 106 patients. The first model developed relied on the CellaVision cell classification tool to generate a diagnosis. The testing results displayed were an AUROC of 0.822 in the discovery cohort and 0.739 in the independent prospective validation cohort. The model was further improved. After cell classification training, based on Multiple Instance Learning (MIL), to eliminate the dependence on a cell classifier such as CellaVision, the results were an AUROC of 0.890 in the discovery cohort and 0.743 in the independent prospective validation cohort [34].

An innovative approach for APL diagnosis based on AI was devised by Yan et al., who developed a DL model that recognizes normal and malignant leukocytes through progressive multigranularity training. The technology used in this study, a segmentation-based enhanced residual network, is also a CNN subtype. The instrument was evaluated for multiple hematological malignancies, including 216 APL images from 7 patients (peripheral blood smears), obtaining a precision of 89.34 ± 0.32, a recall of 97.37 ± 1.34, and an AUROC of 0.913 [44]. 

In a study comparing the Techcyte AI web-based tool for blood smear reading with manual microscopy, 124 peripheral blood films were analyzed, including 22 cases of APL. Three versions were analyzed, AI1 (2019), AI2 (2020), and AI3 (2022). The overall performances of the AI instrument improved from AI1 to AI2 and AI3 in cell identification. The performance of blast cell identification on APL smears increased from a sensitivity of 95% and a positive predictive value of 90% obtained from AI1 to a sensitivity of 100% and a positive predictive value of 91% from AI2 and AI3. According to the manual microscopy result, 20 of 22 APL smears were positive for blasts. The specificity for blast detection and negative predictive value were 0%, 0 out of 2 blast-negative APL smears being identified correctly by the AI tool. In comparison to other similar tools, the Techcyte AI tool removed the “other/unidentified” section of grouping blood cells in AI2 and AI3 versions [45].

Moreover, another method to rapidly identify APL was proposed by Xiao and collaborators who used the CNN-based software CELLSEE to screen for this AML subtype. Their application used over 15,000 bone marrow aspirate smear samples collected from 83 APL patients and 118 controls, which were analyzed using Python at different magnifications. Accuracy was 98% for 10× magnification and 99% for 100× magnification when the conventional deep network ResNet50 was used as the backbone. Overall, the recall was 100% when both magnifications were applied [46].

In addition to reading patient-derived images and establishing diagnoses, AI can aid APL diagnosis by generating images that can be used to train diagnostic tools and morphologists. A system that falls into this category is SyntheticCellGAN, developed, trained, and validated by Barrera et al. This tool was able to generate high-resolution images of normal leukocytes, but also various atypical promyelocytes, highly valuable for APL diagnosis. Pathologists recognized the AI-generated atypical promyelocytes with 100% accuracy. With the ALNet model, the true positive rate obtained for atypical promyelocytes was 91% [47].

### 3.2. Diagnostic Tools Based on Routine Biological Parameters

Apart from the already available specialized morphological tools to assess and grade the patients with cases highly probable of belonging to the category of APL, there is also a constant search to extract more from the data that many centers routinely collect from patients admitted to hospitals. 

Under this category falls also the work of Cheli et al., who proposed in 2022 a tool meant to leverage AI power and to create a tool for early detection of APL based upon routine biological parameters. The tool named Artificial Intelligence for Promyelocytic Leukemia (AIPL) resulted from a training performed on cohort 1 (n = 222)—APL patients (n = 76), non-APL AML (n = 146), and was validated on other 4 cohorts [cohort 2 (n = 44)—APL patients (n = 15)], [cohort 3 (n = 258)—APL patients (n = 46)], [cohort 4 (n = 63)—APL patients (n = 32)], [cohort 5 (n = 50)—APL patients (n = 10)], with a merged validation sample size of n = 415, AUROC = 0.96. The classification algorithm used for the development of this tool was based on XGBoost, as it achieved the maximum performances in the selection phase and the parameters required for the APL tool consisted of: age, white blood cells (absolute value), lymphocytes (% of total leukocytes), neutrophil count (absolute value), mean corpuscular volume (MCV), mean corpuscular hemoglobin concentration (MCHC), prothrombin time ratio (INR), and fibrinogen concentration. In addition to the potential flagging of an APL case, the tool was designed to offer three confidence score categories, high confidence score (>99%), intermediate confidence score (85–99%), and low confidence score (<85%). Even if the described results were impressive in terms of accuracy, the authors concluded that while using the AIPL tool for screening, the cases with a high confidence score (>99%) had no false positive instances while evaluating differential diagnoses [48].

Liao et al. proposed another promising method of incorporating AI into the routine screening of the patients’ laboratory tests in order to determine which might be at a high risk of having APL. The tool this group proposes was meant to address an existing population that fails to be flagged under the International Society of Laboratory Hematology (ISLH) guidelines for morphological review of the blood smears: abnormal white blood cell (WBC) count and an immature granulocyte (IG) flag. The proposed tool was based on ResNet-18-CNN architecture for scattergram quantitative mapping of the complete blood count (CBC) with a differential. The CNN-based model was trained on a set of scattergrams (n = 320 from 51 APL patients, n = 510 from 105 non-APL AML patients, n = 320 from 320 healthy controls) with an external validation subset (n = 56 from 31 APL patients, n = 56 from 55 non-APL AML patients, n = 64 from 64 healthy controls). On average, the CNN-based model displayed promising results, with an AUROC > 0.99 on the subsequent runs of the analysis. The authors described it as a potential new step of a revised workflow to early detect APL based on a supplementary analysis of the CBC scattergram by the CNN-based model [49]. 

Another model of AI usage in the pre-morphological step of CBC evaluation was proposed by Haider and his colleagues as early as 2020. Their research paper was centered on a multiparameter analysis conducted using modern CBC analyzers, with a particular emphasis on the features which investigate CBC research parameters or cell population data (CPD), such as the immature platelet fraction (IPF), DNA/RNA content-related neutrophils (NE-SFL) and absolute platelet count [50]. The proposal involved a “digital signature” as it was referred to, composed of morphological data and immature fractions-related parameters. For the AI component, artificial neural network (ANN) predictive modeling has been selected to interpret heatmap-like data and parameters from principal component analysis (PCA). The research involved 1577 cases with ISLH-compliant criteria (abnormal WBC and ≥2% IG/abnormal WBC) and 96 cases involving APL patients. Subgroup analysis based on the presence of APL returned an AUROC of 0.789 for the model [50].

While the previous tools were focused on the broad screening of patients to identify possible cases of acute leukemia, other groups centered their work on solutions that would speed up the process of identifying the main acute leukemia types—AML, ALL, and APL. The proposed tool did not involve any imaging techniques, but rather relied on the interaction of the disease itself with the human body, in terms of blood cell production, changes in homeostasis/metabolism, and hemostasis. Alcazer and his team recruited data from six independent French university centers and validated using AI and ML techniques (XGBoost) a prediction model based on 10 routine parameters, i.e., age, MCV, MCHC, absolute number of platelets, absolute number of lymphocytes, absolute number of monocytes, percentage of monocytes of the WBC, lactate dehydrogenase (LDH) concentrations, prothrombin time (PT), and fibrinogen concentrations. Artificial Intelligence for Prediction of Acute Leukemia (AI-PAL) emerged from the training of the algorithm on n = 477 datasets, tested on n = 202 instances, and externally validated on a cohort of n = 731. The reported results for the APL subclassification listed an AUROC of 0.97 (95% CI 0.95–0.99), with final accuracy estimates after algorithm optimization for APL predicted as high as 99.7%. However, the authors admitted that the training data have included cases previously flagged as acute leukemia by prior screening, restricting AI-PAL usage to differentiating between AL subtypes. Although this model’s behavior as a screening tool in the general population remained unknown, there were prospects of it being integrated into other ML models that could leverage its accuracy and fast sorting of cases [51].

### 3.3. Diagnostic Tools Based on Multiparameter Flow-Cytometry

While the morphological review of the peripheral blood and bone marrow aspirate smears remains the foundation for a timely diagnosis of acute leukemia, being the standard pivotal point in APL diagnostic work-up, multiparameter flow-cytometry (MFC) data complements the microscopic examination to confirm certain suspected cases. Other defining parameters such as molecular data, i.e., the identification of the *PML::RARA* transcript, usually fall into a 3-day turnaround window, hence the need for MFC analysis to confirm an APL diagnosis sooner. The greatest challenge in the clinic remains the hypogranular variant of APL which might be subjected to lower detection rates [8,9,10]. 

Having set the stage for MFC use in APL, various research groups have proposed pairing MFC with AI and ML algorithms to generate better diagnostic tools for APL. The most recent work published in 2024 by Cox et al. investigated the use of graph neural network (GNN) pipelines. MFC data from a cohort of 68 subjects (27 APL and 41 non-APL AML cases) was mapped using four physical parameters, i.e., forward scatter (FSC) area (FSC-A) and height (FSC-H), side scatter (SSC) area (SSC-A) and height (SSC-H), and 6 fluorescent parameters (CD15, CD33, CD34, HLA-DR, CD117, and CD45). The obtained data were used to construct graphs for each MFC sample via an unsupervised graphics processing unit (GPU) algorithm (PhenoRaft), refined by KNN (K-nearest neighbors) and Jacard similarity coefficient, finally training the GNN-based model to spot disease patterns. The reported accuracy in the testing phase depicted an impressive value of 100%. The model performed well even in samples with low blast percentage (not a common feature of APL) and in the hypogranular variants [52].

A potentially groundbreaking study was published by Monaghan et al. (2022), who proposed an ML-based approach to differentiate acute leukemia from non-neoplastic cytopenia, as well as to discriminate between acute leukemia subtypes. The motivation for this tool resided in the labor-intensive process of manually revising MFC data by laboratory staff, which required a true resource workforce, in addition to its impact on the timespan for the availability of laboratory reports. The dataset used included 531 patients evaluated for non-neoplastic cytopenias/acute leukemia (n = 32 APL; n = 200 non-APL AML; n = 131 ALL; n = 168 non-neoplastic cytopenias). The obtained MFC parameters were used for unsupervised learning by a gaussian mixture model (GMM), followed by a supervised support vector machine classification using Fischer kernel methods. The ML model elicited an AUROC of 0.995 for the full 37 MFC parameter panel and an AUROC of 0.983 for the 3 MFC parameter set, augmenting the power of ML in reducing the need for input data once the machine model is trained. The most powerful weight of the analysis was attributed to the light scatter properties [53]. 

The pioneering work in this field stemmed from the findings of Azad et al., who published a paper as early as 2016 which provided the concept of MFC parameter groups. Their investigation involved collections of MFC data, with each sample being broken into clusters and further organized in meta-clusters in order to obtain a template tree. This hierarchical structure was further used in conjunction with a mixed edge cover (MEC) algorithm for cluster matching and a scoring function was designed. The algorithm was shared by Azad and his colleagues under the name of the R package flowMatch. This template tree allowed the research team to subdivide the AML samples into 13 immunophenotypically distinct categories, amongst which APL was also individualized. The AML dataset used comprised 43 AML patients and 316 healthy controls [54].

### 3.4. Multi-Omics and Machine Learning Applications in APL Assessment

In the field of multi-omics, there are still interactions not fully explored, prompting the development of variable inducible APL cell line models (e.g., U937-PR9) to study the *PML::RARA* gene fusion and its non-canonical pathways. Villiers et al. revealed that 80% of *PML::RARA* binding sites are non-promoter regions, evoking further investigation using advanced techniques like Cut&Run and ATAC-seq. Despite these efforts, transcription factor analysis showed variable transcriptional responses and complex patterns not easily documented through general motif comparison [55].

Based on the unexplored non-promoter regions binding to *PML::RARA*, Villiers and his research team began the search for a solution for exploring these specific pathways. After obtaining the raw data of the experiment, they classified interactions between the transcription factors and the *PML::RARA* transcript as either gained interactions or lost interactions (up-/down*regulated or no change). Transcription factor binding sites (TFBSs) were mapped to document the interaction with the *PML::RARA* gene fusion, obtaining a 436 motif database. eXtreme Gradient Boosting (XGBoost) model was used for binary classification and to predict the regulatory category of a sample, based on the TFBSs displayed, trained using 80% of the samples (with AUROC score of 0,79 obtained on one vs. one pairing method). However, one vs. all comparisons displayed lower AUROC scores (0.55 to 0.70) calling for a finer approach towards unique transcription factor binding site (TFBS) signature identification. Villiers et al. introduced an additional step in the designed decisional scheme, namely the SHAP (SHapley Additive exPlanations) algorithm, a process that conferred weighted contributions of the predictive factors for each transcription factor binding site. Furthermore, the next step involved visualizing the data through t-SNE (t-distributed Stochastic Neighbor Embedding), which highlighted clusters of non-overlapping transcription/interaction categories, defined by the presence or absence of various TFBSs. The authors concluded that it is still premature to use this model to pinpoint certain motifs and their unique relationship to a certain regulatory cluster, but the clear separation of these clusters reinforces the concept of genetic regulatory programs comprising a particular expression heatmap [55]. The exploratory ML algorithm was termed REBEL (Regulatory Element Behavior Extraction Learning) and was intended to serve as a tool focused on integrating multi-omics datasets to explore the regulatory activities of transcription factors and their interactions. This approach provided valuable insights into complex regulatory pathways which were uncovered through the use of refined ML techniques that facilitated TFBS pattern recognition and pairing with the involved regulatory pathways, in an attempt to clarify biological aspects of the APL phenotype [55]. 

In addition, Hu et al. investigated the potential uses of a platform entitled APAview to assess the roles of alternative polyadenylation (APA) dysregulation in APL and other blood cancers. In total, they examined 155 genes related to alternative polyadenylation, revealing that the *JAK1* gene exhibited two APA sites. Moreover, its expression was negatively associated with the expression of the PDUI gene and positively associated with the expression of several other genes, including *PTPN11*, *SOCS5*, *MDM2*, *GRB2*, *STAT1,* and *STAT3*, which are involved in the differentiation of APL cells. Therefore, the decreased expression of *JAK1* and of the aforementioned genes might play a role in APL onset [56].

Thrun and collaborators also employed bioinformatics to quantify surface molecules in various AML subtypes, including APL, using a combined approach based on ABC and Bayesian analysis, respectively. Their assessment made use of two gene expression datasets from 431 AML cases of whom 30 were APL, as well as healthy controls. RNA sequencing and microarray data revealed a distinctive hallmark of APL in comparison with other AML subtypes, i.e., an elevated expression of the CD339 gene [57].

### 3.5. Qualitative Analysis of the Performance Metrics of AI Models Used for APL Assessment

Table 2 summarizes the main characteristics of the AI models with applications in the diagnosis and management of APL, highlighting the input data required by each tool to generate a prediction or a result relevant to APL clinical workflows and the performance characteristics obtained from the testing phase described in each study.

### 3.6. Quantitative Analysis of the Performance Metrics of AI Models Used for APL Assessment

Table 3 reports the performance metrics of the AI models analyzed in the current review.

### 3.7. Quantitative Analysis of the Performance Metrics of AI Models That Used Bone Marrow Aspirate Smears for APL Assessment

Overall, the AI models for which the input data consisted of bone marrow aspirate smears displayed a mean sensitivity (TPR) of 0.958775. Amongst the three types of input data assessed, it displayed the highest mean sensitivity, reinforcing the AI models’ strong ability to detect APL. The mean specificity (TNR) of the AI models was 0.935, slightly lower than the AI models that employed peripheral blood smears. However, these findings still indicate reliable performances. The models’ mean accuracy was 0.9295, ranging from 0.87 to 0.99 and depicting a relatively stable performance. The mean AUROC was 0.9465, demonstrating high reliability in classifying APL cases.

Table 4 summarizes the performance metrics of the AI models which used bone marrow smears as input data.

### 3.8. Quantitative Analysis of the Performance Metrics of AI Models That Used Peripheral Blood Smears for APL Assessment

Overall, the AI models for which the input data consisted of peripheral blood smears displayed a mean sensitivity (TPR) of 0.92366, indicating the combined ability of the proposed models to correctly identify APL cases. Its TPR ranged from 0.8 to 0.9919, revealing a slight variability in APL detection accuracy. These AI models exhibited a mean specificity (TNR) of 0.9765, highlighting their elevated ability to correctly identify non-APL cases. It is worth mentioning that these AI models displayed a low standard deviation, pointing out that the final output of the models investigated exhibited consistency. The mean accuracy of these AI models was 0.9428, demonstrating the overall correctness of these models. The minimal observed accuracy was 0.87 and the maximal observed accuracy was 0.9954, stressing that there is a degree of variability in the returned results. The mean AUROC of these AI models was 0.9563, reflecting excellent performance metrics that warrant further consideration for optical methods when there is a need to discriminate between APL and non-APL cases based on peripheral blood smears.

Table 5 summarizes the performance metrics of the AI models which used peripheral blood smears as input data.

### 3.9. Quantitative Analysis of the Performance Metrics of AI Models That Used Other Parameters for APL Assessment

The AI models involving biochemical parameters, clusters of differentiation (CDs), or transcription factor motifs exhibited a mean sensitivity (TPR) of 0.937766667, a remarkably high value that indicates good detection capabilities. Their mean specificity (TNR) was 0.980467, the highest among the three data types, demonstrating excellent performance in ruling out non-APL cases. Their mean accuracy was 0.969067, the highest accuracy among the three major AI-aided investigation methods, indicating the models’ strong performance. Their mean AUROC was 0.9175, slightly lower versus the other data types, but still suggestive of good classification performance.

Table 6 summarizes the performance metrics of the AI models which used other biomarkers as input data.

Table 7 summarizes the mean values of the performance metrics calculated for the AI models proposed in the APL assessment.

The AI tools based on peripheral blood smears displayed the highest accuracy and AUROC value, revealing solid performances in terms of sensitivity and specificity. Therefore, these models could be considered for further clinical testing and integration within APL assessment. The highest sensitivity (TPR) was obtained using bone marrow aspirate samples, but with lower accuracy and AUROC values which prompts for improvements in the ANN models used to assess these biological samples. Furthermore, the most specific results were obtained during the experiments which assessed other biomarkers (highest TNR and accuracy), underscoring the particular changes in the homeostasis of the human body and the pattern recognition ease that a trained GNN could leverage.

## 4. Discussions, Future Perspectives and Conclusions

Herein, by means of a comprehensive review, we examined for the first time the potential applications of AI and its branches, e.g., ML, DL, CNNs, and others, in the diagnosis and management of APL. In the context of a pronounced emergency character of APL cases and the importance of prompt treatment initiation, obtaining rapid and accurate diagnosis in this AML subtype is crucial, leading to a need that might be fulfilled with the help of AI. Thus, AI might aid physicians in establishing a rapid and accurate diagnosis of APL and in the decision to commence treatment (Figure 3).

Manual microscopic evaluation of peripheral blood and bone marrow aspirate films is a central step in the diagnosis of hematological malignancies, justifying why the majority of AI-based tools for APL were focused on automatic reading and interpretation of smears. While being highly valuable in patient management, manual microscopy relies on trained personnel availability, can be influenced by fatigue and overload, and is time-consuming. These specific limitations can be addressed and resolved with the aid of AI, thus improving accuracy, and rapidity and reducing morphologists’ overload [39]. For example, the AI model developed by Eckardt et al. can establish an APL diagnosis in less than 1 min [40].

The most performant AI technology for image classification was DL. Furthermore, relatively recently, the errors in classifications were drastically decreased through the development of CNN, one of the three DL approaches [41]. CNNs have displayed elevated accuracy in the recognition and classification of both normal mature leukocytes and their precursors, leading to multiple opportunities in the diagnostic work-up of hematological malignancies [43].

Most of the methods described in the existing literature involved ANN models to detect APL. Experiments varied in terms of cohort sizes and in terms of the methods evaluated. A significant proportion of studies have been conducted on preexisting optical evaluation techniques which involve capturing blood film images and further data manipulation. The most explored experimental design uses CNNs which screen the optical image acquired. Even if the datasets from each reporting center were small, most of the studies reported external validation with cross-sets provided by other research centers. A limitation worth mentioning was the high variation in data collection as pointed out by Barrera et al., given that staining and image capture might vary across the commercially available staining and scanning laboratory devices [47]. Lincz and his colleagues described how various iterations of AI tools achieved a remarkable 98% rate of sensibility in identifying blast cells, sometimes at the expense of specificity which decreased to 12% in AI3, an acceptable trade-off for screening tools [45].

The graph neural network (GNN) proved to be most suitable in more complex scenarios where MFC has been explored as the source of data for the AI models. Cox et al. analyzed a small sample of patients (n = 27 APL) with an AI model which displayed an accuracy of 100%. This is a possible scenario as this method is not hindered by staining and acquisition factors [52]. This result is encouraging for unsupervised MFC algorithms, although the real use of MFC in the screening setting is limited as immunophenotyping still involves sample preparation steps which require human assistance. However, we cannot fail to recognize that such an AI component might benefit the clinical laboratory due to its precision, complementing a manual cytometry report. Nonetheless, another paper documented the possibility of reducing the analyzed parameters from 37 to just 3 (FSC-A, SSC-H, CD117), with similar AUROC performance [53]. While not yet a widespread quick identification method, MFC-paired machine learning GNN structures might be integrated as aiding systems and also the correct marker selection might help reduce the cost of the procedure when processed through a previously trained ML.

The potential applications of AI in the diagnosis, evaluation, and management of APL would benefit not only high-income economies but also low- and middle-income countries where access to advanced laboratory instruments and techniques, i.e., molecular diagnostics, is limited or non-existent. AI, ML, and DL can rapidly identify atypical promyelocytes and signal physicians that a patient might suffer from APL leading to a more urgent smear review and referral to a specialized hematology department, especially since there is a need to reduce the rates of very early and early deaths in this malignancy [51]. Very early death, i.e., exitus before therapy initiation in APL, can occur in approximately 5% of cases, particularly in subjects who present with disseminated intravascular coagulation, elevated creatinine, or require mechanical ventilation. Early death during the first week after APL diagnosis can occur in 12.5% of cases and seems to be predicted in particular by the increase in creatinine concentrations (OR: 21.4, 95% CI 2.2–205.8, *p* = 0.008) [59]. Thus, AI and related computer technologies might help identify patients at risk of very early and early death in APL.

Moreover, the use of AI might aid in the reduction in reagent use and render APL identification procedures less time-consuming, e.g., MFC can establish a diagnosis of APL using only three parameters as suggested by Monaghan et al. [53]. In addition, multi-omics enriched by AI might help understand the complex interactions between APL signaling pathways and the clinical profile and outcome of patients who are diagnosed with this malignancy.

Despite the excellent accuracy of AI in the evaluation of APL cases, with most AUROC values surpassing 90%, we must stress that AI can only aid physicians in establishing an APL diagnosis and that computers cannot replace healthcare workers, particularly as their training requires the use of the human mind.

The road to the implementation of AI tools in APL diagnosis in healthcare facilities is long and challenging. While the majority of the studies reported excellent performance characteristics, most of these tools are in the early stages of development and testing. With few exceptions, the majority of the AI tools with potential applications in APL management were tested on small datasets, lacked external validation, and were evaluated mostly outside of the clinical settings. While some models were trained and tested with many thousands of images, these images were obtained from a relatively small group of APL patients. With APL diagnosis being an emergency and implying immediate therapeutic and supportive interventions, the safety of implementing AI tools in clinical practice with this purpose must be thoroughly evaluated. The above-described studies reported promising results, but further research is required to bring technological progress closer to the hematology clinics. Another limitation is that flow-cytometry and omics-based techniques to diagnose APL are not widely available worldwide, with low-income economies encountering potential barriers in implementing AI-guided APL screening. Moreover, in order to train AI software, there is a need for adequate preanalytical processing of biological samples as well as experts in APL and bioinformaticians. It is also noteworthy to mention that several ML systems used in APL diagnostics use self-supervised and weakly supervised approaches that are prone to errors as previously demonstrated elsewhere [60].

From a statistical perspective, the types of input data and the subsequent ANN models used proved to have their own strengths, worthy to be integrated as complementary tools in the diagnostic process of APL. Peripheral blood smears-based AI assessment seemed to offer a balanced approach, while bone marrow aspirate smears-guided AI instruments resulted in higher sensitivity rates. ANN-conducted interpretations of other laboratory data ensured elevated accuracy and specificity. Further research is needed to verify whether combinations of these AI methods might improve the accuracy of APL diagnosis prior to a standard hematopathology evaluation.

The current narrative review indicates that efforts have been made to develop several AI models with applications in the diagnosis and case management of APL. While most of them might seem to perform similarly, a comparison is challenging in the context of the varying stages of development of the tools, differences in evaluation approaches, and the diversity of input data and analytical methods used to generate predictions or results. In conclusion, the results of our analysis suggest that AI can emerge as a relevant tool in the evaluation of APL cases and potentially contribute to more rapid screening and identification of this hematological emergency. AI models based on cytomorphology displayed the highest specificity, accuracy, and AUROC for peripheral blood smears and the highest sensitivity for bone marrow aspirate smears. AI methods based on other biomarkers exhibited superior specificity and accuracy but lower sensitivity and AUROC values.

Future research should focus on the selection of the best AI method to diagnose APL, reduce the rate of false negatives and false positives, as well as investigate the potential benefits of multi-modal data analysis systems which could be relevant to understanding how to combine different diagnostic inputs, e.g., complete blood count data, morphology, flow-cytometry, omics, and others. Moreover, international collaboration is needed to create a large, freely, and widely available APL dataset that could serve to validate AI methods. Future investigations should explore the use of other CBC-derived indices, biochemical markers, hemostasis screening tests, and cytomorphology in the diagnosis of APL, as well as in the prediction of its complications, for example, to predict the risk of disseminated intravascular coagulation, infections or treatment-related side effects. This would be of particular importance in resource-limited settings where MFC, FISH, or molecular biology are not widely available.

## Figures and Tables

**Figure 1 jcm-14-01670-f001:**
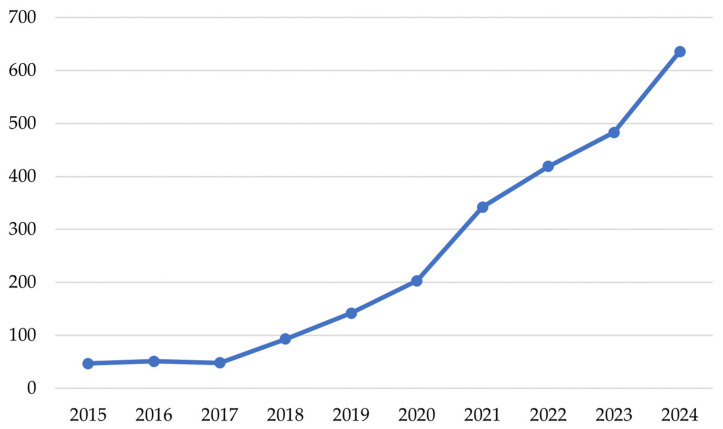
Publication trends on the use of AI in malignant hematology. The number of manuscripts indexed in PubMed/MEDLINE [12] and related to this topic has steadily increased since the year 2017.

**Figure 2 jcm-14-01670-f002:**
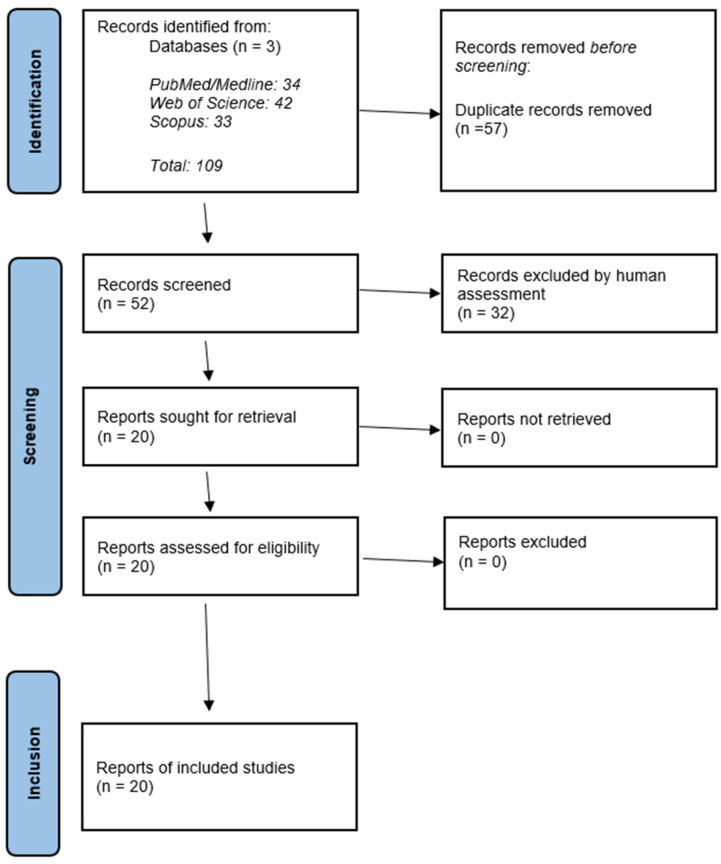
PRISMA 2020 flow diagram [35] for the current review which investigated the use of AI in APL. AI, artificial intelligence. APL, acute promyelocytic leukemia.

**Figure 3 jcm-14-01670-f003:**
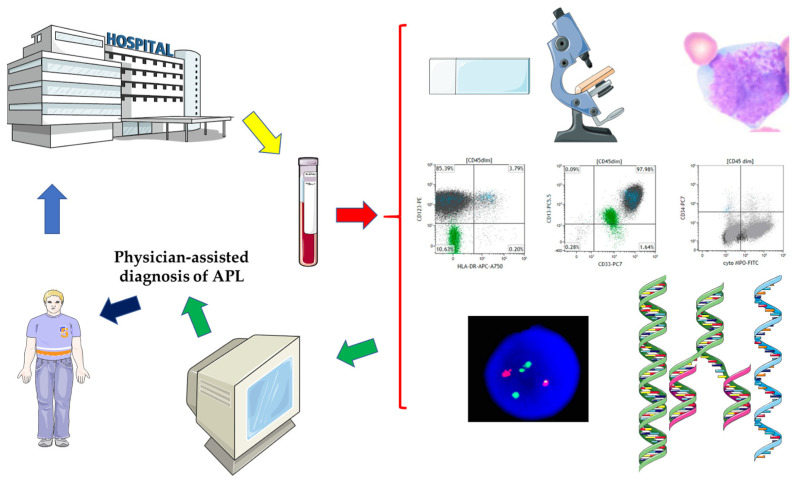
Proposed algorithm for physician-assisted APL diagnosis using AI tools. **Legend:** A patient presents to the hospital and blood samples are collected and subjected to different laboratory tests. In comparison to the traditional diagnosis workflow, the new algorithm involves, after sample collection, an AI-assisted analysis of data: peripheral blood and bone marrow aspirate smears, scattergrams from flow-cytometry analysis, FISH analysis for t(15;17) detection and molecular biology to confirm the presence of the *PML::RARA* gene fusion. AI enables a faster and more accurate diagnosis and immediate life-saving treatment initiation by hematologists. During conception and drawing, medical illustrations from SMART and peripheral blood smears, flow-cytometry scatter plots, and fluorescence in situ hybridization images from George GV et al., Genes 2025, DOI: 10.3390/genes16010007 [58], were used. Please refer to the Acknowledgments.

**Table 1 jcm-14-01670-t001:** Training and testing datasets for MILLIE (extracted and synthesized from [39].

Datasets for Model Training and Testing	Results
PBS: 30 APL samples and 40 other samples (normal PBS, other AML subtypes); random split of the samples—three-fold cross-validation; 2/3 for training and 1/3 for testing	AUROC (sample classification): 0.935 ± 0.036 (diagnostic labels)
Single-cell images (PBS) of 611 promyelocytes and 3000 other myeloid and normal WBC (testing set)	AUROC: 0.88 (discrimination of promyelocytes from other types of WBC)
Further training and testing—33 APL samples and 72 AML samples (PBS)—random three-fold cross-validation	AUROC: 0.96 ± 0.02
BMS from 236 healthy individuals and 1095 AML patients (43 with APL)—training 3/4 and testing 1/4, randomly split; four-fold cross-validation	AUROC (average): 0.99
Single-cell images (BMS) of 309 promyelocytes, 718 myeloblasts, and 262 normal WBC (testing set)	AUROC (promyelocytes): 0.895; most promyelocytes (78%) are classified correctly

Legend: AML, acute myeloid leukemia. APL, acute promyelocytic leukemia. AUROC, area under the receiver operating characteristic curve. BMS, bone marrow aspirate smear. PBS, peripheral blood smear. WBC, white blood cells.

**Table 2 jcm-14-01670-t002:** The main characteristics of the studies that proposed AI models for APL assessment.

Study	Tool	Input Data *	Results **
Manescu, 2023 [39]	Multiple Instance Learning for Leukocyte Identification (MILLIE)	PBS and BMS	AUROC (PBS): 0.94 ± 0.04 AUROC (BMS): 0.99 ± 0.01
Eckardt, 2024 [40]	Custom AI tool, unnamed	BMS	AUROC: 0.8575–0.9585 (varies between the comparison groups)
Ouyang, 2020 [41]	Custom AI tool, unnamed	BMS	Average precision: 62.5% Average recall: 84.1%
Qiao, 2021 [42]	Custom AI tool, unnamed	PBS	AUROC: 0.9977 ± 0.0003 or 0.9914 ± 0.0026 Precision: 97.65% or 99.2% (depending on the dataset)
Boldu, 2021 [43]	ALNet	PBS	Correct diagnosis prediction for all APL cases
Sidhom, 2021 [34]	Custom AI tool, unnamed	PBS	AUROC (discovery cohort): 0.890 AUROC (independent prospective validation cohort): 0.743
Yan, 2024 [44]	Custom AI tool, unnamed	PBS	Precision: 89.34 ± 0.32 Recall: 97.37 ± 1.34 AUROC: 0.913
Lincz, 2023 [45]	Techcyte AI tool	PBS	Blast detection performances in APL: Sensitivity 100% Specificity: 0% Positive predictive value: 91% Negative predictive value: 0%
Barrera, 2023 [47]	SyntheticCellGAN	Images generated from iterations of PBS	Accuracy (atypical promyelocytes): 100% Accuracy (pathologist’s interpretation): 91% Positive predictive value (ALNet prediction)
Xiao, 2024 [46]	CELLSEE AI-powered APL morphological diagnostic system	BMS	AUROC: 0.9708
Cheli, 2022 [48]	AI for APL	Biological parameters: age, WBC, Ly (% of WBC), NE (absolute value), MCV, MCHC, INR, fibrinogen concentrations	AUROC: 0.96
Liao, 2023 [49]	ResNet-18-CNN architecture for scattergram mapping	CBC scattergrams	AUROC: >0.99
Haider, 2022 [50]	Cell population data-driven ANN predictive modeling	CBC research parameters/cell population data	AUROC: 0.789
Alcazer, 2024 [51]	AI prediction of acute leukemia (AI-PAL)	Biological parameters: age, MCV, MCHC, PLT (absolute number), LY (absolute number), MO (absolute number), MO% (% of WBCs), LDH, PT, fibrinogen concentration	AUROC: 0.97
Cox, 2024 [52]	GNN pipeline using MFC data	MFC data: 4 physical parameters (FSC-A, FSC-H, SSC-A, SSC-H), 6 fluorescent parameters (CD15, CD33, CD34, HLA-DR, CD117, CD45)	AUROC: 1
Monaghan, 2022 [53]	Machine learning to classify acute leukemias and distinction from nonneoplastic cytopenias using GMM and Fisher kernel methods on MFC data	MFC data using 37 FC parameters MFC (3 parameters)	AUROC (37 FC parameters): 0.995 AUROC (3 MFC parameters): 0.983
Azad, 2016 [54]	FlowMatch	MFC data	N/A
Villiers, 2023 [55]	Regulatory Element Behavior Extraction Learning (REBEL)	Transcription factor motif datasets, previously distilled by machine learning algorithms	AUROC: 0.51–0.64
Thrun, 2022 [57]	Bayesian and ABC Analysis	MFC parameters (CD paired analysis APL vs. non-APL sample structure via microarrays)	N/A—exploratory results identifying particular novel CD markers for APL (e.g., CD339)
Hu, 2022 [56]	APAview—web-based platform for alternative polyadenylation analyses in hematological cancers	APA sites labeling and quantifying APA usages	N/A—exploratory results analyzing various gene pathways involvement in hematological malignancies (e.g., *JAK1* and *STAT1*, *STAT3*, *GRB2*, *SOCS5*, *PTPN11*, and *MDM2*—in APL)

Legend: AI, artificial intelligence. APL, acute promyelocytic leukemia. AUROC, area under the receiver operating characteristic. APA, alternative polyadenylation. BMS, bone marrow aspirate smear. CBC, complete blood count. CD, cluster of differentiation antigens. FC, flow-cytometry. GNN, graph neural network. GMM, gaussian mixture model. INR, international normalized ratio. Ly, lymphocytes. MCHC, mean corpuscular hemoglobin concentration. MCV, mean corpuscular volume. MFC, multiparameter flow-cytometry. NE, neutrophils. PBS, peripheral blood smear. * to generate predictions (i.e., diagnosis, prognosis, response to treatment, etc.); ** only those applied to APL.

**Table 3 jcm-14-01670-t003:** Summarization of the performance metrics of the AI models presented in the analyzed studies.

No.	Author	Sample Type	External Control	Sensitivity (TPR)	Specificity (TNR)	Accuracy	AUROC
1	Xiao, 2024 [46]	BMS	Yes	0.9080	0.8500	0.9380	0.9800
2	Manescu, 2023 [39]	BMS	Yes	1.0000	0.9800	0.9900	0.9900
3	Eckardt, 2024 [40]	BMS	No	0.9671	0.9400	0.8700	0.9585
4	Ouyang, 2020 [41]	BMS	No	0.9600	0.9700	0.9200	0.8575
5	Manescu, 2023 [40]	PBS	Yes	0.8000	0.9400	0.8700	0.9400
6	Qiao, 2021 [42]	PBS	Yes	0.9919	0.9988	0.9954	0.9585
7	Boldu, 2021 [43]	PBS	Yes	0.9530	1.0000	0.9470	0.9800
8	Sidhom, 2021 [34]	PBS	Yes	0.8000	0.9000	N/A	0.7430
9	Yan, 2024 [44]	PBS	Yes	0.8934	0.9737	0.9318	0.9130
10	Lincz, 2023 [45]	PBS	No	AI1: 0.97, AI2: 0.98, AI3: 1	AI1: 0.24, AI2: 0.14, AI3: 0.12	N/A	N/A
11	Barrera, 2023 [47]	PBS (processed images)	No	0.9800	0.9700	0.9700	0.9900
12	Cheli, 2022 [48]	Biological parameters	Yes	0.8456	0.9398	0.9614	0.9600
13	Alcazer, 2024 [51]	Biological parameters	Yes	0.9610	0.9970	0.9610	0.9600
14	Liao, 2023 [49]	CBC	Yes	0.9500	1.0000	0.9600	0.9900
15	Haider, 2022 [50]	CBC	Yes	N/A	N/A	0.995 (high score), 0.85 (intermediate score), 0.68 (low score)	0.9600
16	Thrun, 2022 [57]	CDs	N/A	N/A	N/A	N/A	N/A
17	Cox, 2024 [52]	CDs	Yes	1.0000	1.0000	1.0000	1.0000
18	Monaghan, 2022 [53]	CDs	Yes	0.8750	0.9560	0.9420	0.9550
19	Azad, 2016 [54]	CDs	No	0.8750	1.0000	N/A	N/A
20	Villiers, 2023 [55]	Transcription factor motifs	No	0.9950	0.9900	0.9900	0.6400
21	Hu, 2022 [56]	APA	N/A	N/A	N/A	N/A	N/A

Legend: TPR, true positive rate. TNR, true negative rate. AUROC, area under the receiver operating characteristic. N/A, not applicable. BMS, bone marrow aspirate smears. PBS, peripheral blood smears. CBC, complete blood count. CD, cluster of differentiation.

**Table 4 jcm-14-01670-t004:** Performance metrics of AI models that used bone marrow smears as input data for APL assessment.

No.	Author	Sample Type	External Control	Sensitivity (TPR)	Specificity (TNR)	Accuracy	AUROC
1	Xiao, 2024 [46]	BMS	Yes	0.9080	0.8500	0.9380	0.9800
2	Manescu, 2023 [39]	BMS	Yes	1.0000	0.9800	0.9900	0.9900
3	Eckardt, 2024 [40]	BMS	No	0.9671	0.9400	0.8700	0.9585
4	Ouyang, 2020 [41]	BMS	No	0.9600	0.9700	0.9200	0.8575

Legend: TPR, true positive rate. TNR, true negative rate. AUROC, area under the receiver operating characteristic. BMS, bone marrow aspirate smears.

**Table 5 jcm-14-01670-t005:** Performance metrics of AI models that used peripheral blood smears as input data for APL assessment.

No.	Author	Sample Type	External Control	Sensitivity (TPR)	Specificity (TNR)	Accuracy	AUROC
1	Manescu, 2023 [39]	PBS	Yes	0.8000	0.9400	0.8700	0.9400
2	Qiao, 2021 [46]	PBS	Yes	0.9919	0.9988	0.9954	0.9585
3	Boldu, 2021 [43]	PBS	Yes	0.9530	1.0000	0.9470	0.9800
4	Yan, 2024 [44]	PBS	Yes	0.8934	0.9737	0.9318	0.9130
5	Barrera, 2023 [47]	PBS	No	0.9800	0.9700	0.9700	0.9900

Legend: TPR, true positive rate. TNR, true negative rate. AUROC, area under the receiver operating characteristic. PBS, peripheral blood smears. Note: Sidhom, 2021 [34] and Lincz, 2023 [45]—data censored as various variables were not reported in the reviewed papers regarding one or more elements (AUROC, TPR, TNR, accuracy).

**Table 6 jcm-14-01670-t006:** Performance metrics of the AI models which used other biomarkers as input data.

No.	Author	Sample Type	External Control	Sensitivity (TPR)	Specificity (TNR)	Accuracy	AUROC
1	Cheli, 2022 [48]	Biological parameters	Yes	0.8456	0.9398	0.9614	0.9600
2	Alcazer, 2024 [51]	Biological parameters	Yes	0.9610	0.9970	0.9610	0.9600
3	Liao, 2023 [49]	CBC	Yes	0.9500	1.0000	0.9600	0.9900
4	Cox, 2024 [52]	CDs	Yes	1.0000	1.0000	1.0000	1.0000
5	Monaghan, 2022 [53]	CDs	Yes	0.8750	0.9560	0.9420	0.9550
6	Villiers, 2023 [55]	Transcription factor motifs	No	0.9950	0.9900	0.9900	0.6400

Legend: TPR, true positive rate. TNR, true negative rate. AUROC, area under the receiver operating characteristic. CBC, complete blood count. CD, cluster of differentiation. Note: Haider, 2022 [50]; Thrun, 2022 [57]; Hu, 2022 [56], Azad, 2016 [54]—data censored as various variables were not reported in the reviewed papers regarding one or more elements (AUROC, TPR, TNR, Accuracy).

**Table 7 jcm-14-01670-t007:** Mean values of the performance metrics of AI models for APL assessment.

Mean Values for Performance Metrics	PBS	BMS	Other Biomarkers
TPR (Sensitivity)	0.92366	0.958775	0.937766667
TNR (Specificity)	0.9765	0.935	0.980467
Accuracy	0.9428	0.9295	0.969067
AUROC	0.9563	0.9465	0.9175

Legend: TPR, true positive rate. TNR, true negative rate. AUROC, area under the receiver operating characteristic. PBS, peripheral blood smears. BMS, bone marrow aspirate smears.

## Data Availability

Data are available on request from authors.

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
