# Peer review of "Applications of Artificial Intelligence in Acute Promyelocytic Leukemia: An Avenue of Opportunities? A Systematic Review"

_jcm, 2025, doi:10.3390/jcm14051670_

Round 1

Reviewer 1 Report

Comments and Suggestions for Authors

Dear Authors,

I recently had the opportunity to read your manuscript titled “Applications of Artificial Intelligence in Acute Promyelocytic Leukemia: An Avenue of Opportunities?” and I would like to reach out to you to express my comments about your work.

This manuscript reviews the applications of artificial intelligence (AI), machine learning (ML), and deep learning (DL) in the diagnosis, evaluation, and management of acute promyelocytic leukemia (APL). By analyzing 17 studies, it highlights AI-driven tools such as convolutional neural networks (CNNs) and graph neural networks (GNNs) for automated diagnostics and multi-omics approaches for gene regulatory analyses. The review underscores the potential of AI technologies to improve diagnostic accuracy, reduce healthcare disparities, and accelerate clinical workflows, particularly in resource-limited settings.

Nevertheless, here are some possible comments outlining areas that could improve the quality and readability of the manuscript:

Introduction:

1.     Research Aims/Objectives: The research objectives are broadly stated but lack precision. They should be clearly defined and more focused on the specific gaps in applying AI to acute promyelocytic leukemia (APL) management.

2.     Background Context: While the introduction provides general context on APL and AI applications, it fails to sufficiently review prior key studies. Adding more recent, high-impact references on AI in oncology would enhance the rationale.

Methods:

3.     Study Design/Protocol: The methodology for selecting and analyzing the 17 studies is unclear. Specific criteria for inclusion/exclusion should be provided, such as database sources, timeframes, and keywords used in the literature search.

4.     Statistical Analysis: No statistical methodology is described, making it difficult to assess the robustness of the analysis. If performance metrics or comparisons were used to evaluate AI tools, these need to be explicitly mentioned and justified.

5.     Replicability: The lack of detail in the methods section hinders replication. Clarify how AI approaches (e.g., CNNs or GNNs) were assessed and whether external validation data were considered.

Results:

6.     Alignment with Aims: While the results summarize the reviewed studies, they do not directly address how AI improves APL diagnosis and management compared to traditional methods. A clearer linkage to the study’s objectives is needed.

7.     Presentation Clarity: Data presentation is inconsistent, with some tables and figures lacking adequate labeling and explanation. For example, ensure all figures have descriptive captions explaining their relevance to the results.

8.     Comprehensiveness: Some key findings are discussed vaguely or omitted. A systematic comparison of the AI methods (e.g., accuracy, scalability, clinical applicability) should be included.

Discussion:

9.     Key Findings and Interpretation: The discussion highlights AI’s potential but lacks depth in interpreting the key findings within the context of existing literature. Further analysis of why certain AI tools perform better in APL management is needed.

10.  Limitations: While some limitations (e.g., small study sample) are briefly mentioned, others, such as biases in the reviewed studies and generalizability to different populations, are not adequately addressed.

11.  Practical Implications: The broader implications of AI in improving diagnostic workflows and reducing healthcare disparities need further exploration. Propose concrete strategies for implementing these findings in clinical settings.

Conclusions:

12.  Addressing Aims: The conclusions are general and fail to explicitly address the original research aims. Clearly summarize how the study contributes to the understanding of AI applications in APL.

13.  Significance: The implications for future research and clinical practice are not well articulated. Provide actionable recommendations for researchers and clinicians to advance AI integration in oncology.

The manuscript demonstrates significant linguistic issues that need thorough revision. Grammatical inconsistencies, including article misuse and incorrect verb tenses, impede readability. Sentence structures are often verbose, leading to redundancy and lack of clarity. Terminology, while technically accurate, is at times overly complex, making the content less accessible to a broader audience. Additionally, the transitions between sections are abrupt, affecting coherence. A comprehensive professional English editing is necessary to meet the standards of high-impact journals.

Once again, thank you very much for your work. We´ll be waiting for your answers about my comments.

Kindest regards,

Comments on the Quality of English Language

The manuscript demonstrates significant linguistic issues that need thorough revision. Grammatical inconsistencies, including article misuse and incorrect verb tenses, impede readability. Sentence structures are often verbose, leading to redundancy and lack of clarity. Terminology, while technically accurate, is at times overly complex, making the content less accessible to a broader audience. Additionally, the transitions between sections are abrupt, affecting coherence. A comprehensive professional English editing is necessary to meet the standards of high-impact journals.

Author Response

Dear Editor in Chief,

Dear Academic Editor and Guest Editors,

We are very thankful to you and to the peer-reviewers for the pertinent notes; we have carefully read the comments and have revised/completed the manuscript accordingly. Our responses are given in a point-by-point manner below. All the changes to the manuscript are highlighted in yellow. We hope that, in this new form, the manuscript will be suitable for publication in the Journal of Clinical Medicine.

Dear Authors, I recently had the opportunity to read your manuscript titled “Applications of Artificial Intelligence in Acute Promyelocytic Leukemia: An Avenue of Opportunities?” and I would like to reach out to you to express my comments about your work.

This manuscript reviews the applications of artificial intelligence (AI), machine learning (ML), and deep learning (DL) in the diagnosis, evaluation, and management of acute promyelocytic leukemia (APL). By analyzing 17 studies, it highlights AI-driven tools such as convolutional neural networks (CNNs) and graph neural networks (GNNs) for automated diagnostics and multi-omics approaches for gene regulatory analyses. The review underscores the potential of AI technologies to improve diagnostic accuracy, reduce healthcare disparities, and accelerate clinical workflows, particularly in resource-limited settings.

Nevertheless, here are some possible comments outlining areas that could improve the quality and readability of the manuscript:

Introduction:

  1. Research Aims/Objectives: The research objectives are broadly stated but lack precision. They should be clearly defined and more focused on the specific gaps in applying AI to acute promyelocytic leukemia (APL) management.

Response: Thank you for your suggestion. We have updated the manuscript to include a broader explanation of the need for AI screening tools in APL as instructed:

Hematological malignancies rely to a high degree on laboratory techniques which involve the morphological recognition of the abnormal cells and then also refine the results of the preliminary evaluation through more advanced investigations, such as immunophenotyping. Acute leukemias prompt a time sensitive approach in terms of the recognition speed, acute promyelocytic leukemia (APL) eliciting the highest need of prompt treatment initiation, as prompt initiation of all-trans retinoic acid (ATRA) treatment can prevent lethal complications which might arise. This particular feature made researchers turn towards the use of AI in identifying APL cases faster and in a scalable manner. Unlike the traditional route, most of the works encountered in the existing literature sought to leverage convolutional neural networks (CNNs), which are proven to be highly effective in assessing medical imaging data, hence the suitability of this method to process cellular input data obtained from larger cohorts of patients.

The main drivers of these AI models have focused on the scaling capacities, being able to process high volumes of data, which would have required a lot of trained health care professionals for manual reviewing, on the consistency of the results that a CNN model can elicit, being able to recognize even subtle morphological challenges and also, maybe the most important factor, the accessibility of these methods in various areas which lack dedicated laboratory facilities involved in daily hematopathology analyses. Therefore, AI in APL would play a vital role in the screening phase of this pathology, promising to be an efficient tool for screening mass samples and detecting the profiles which might match this differential diagnosis, which is indeed considered a hematological emergency.” 

  1. Background Context: While the introduction provides general context on APL and AI applications, it fails to sufficiently review prior key studies. Adding more recent, high-impact references on AI in oncology would enhance the rationale.

Response: Thank you for your suggestion. We have revised the introduction to include several recent, high-impact references that highlight clinically implemented AI tools in oncology and hematology. We believe these references enhance the context and underscore the relevance of our research within the broader field of AI-driven oncology and hematology.

Methods:

  1. Study Design/Protocol: The methodology for selecting and analyzing the 17 studies is unclear. Specific criteria for inclusion/exclusion should be provided, such as database sources, timeframes, and keywords used in the literature search.

Response: Thank you for your suggestion. We have clarified the methodology of the study, screened two additional databases (Web of Science and Scopus), and added inclusion/exclusion criteria, timeframes and keywords. A PRISMA flowchart was also added to the paper.

  1. Statistical Analysis: No statistical methodology is described, making it difficult to assess the robustness of the analysis. If performance metrics or comparisons were used to evaluate AI tools, these need to be explicitly mentioned and justified.

Response: Thank you for your suggestion. As this is a narrative review, we did not conduct an independent statistical analysis. However, we recognize the importance of explicitly mentioning the performance metrics considered from the studies we reviewed. To address this, we have revised the manuscript Methods section. Additionally, we have clarified that no independent statistical methodology was applied in this review.

  1. Replicability: The lack of detail in the methods section hinders replication. Clarify how AI approaches (e.g., CNNs or GNNs) were assessed and whether external validation data were considered.

Response: Thank you for your suggestion. We have included a descriptive paragraph which explains the AI techniques suggested by each of the studies presented in this review.

Results:

  1. Alignment with Aims: While the results summarize the reviewed studies, they do not directly address how AI improves APL diagnosis and management compared to traditional methods. A clearer linkage to the study’s objectives is needed.

Response: Thank you for your valuable feedback. We have revised both the Methods section and the Results section to better align the study’s aims with the presented content. Specifically, we have clarified the key aspects relevant to our objectives, including details about the AI model itself, the training and testing phases, the datasets used, dataset sizes, and the reported performance characteristics in relation to the development phase. These elements have been explicitly described in the Methods section and systematically extracted from each included article, with the findings presented in the Results section. We believe these revisions strengthen the linkage between our objectives and the study’s findings.

  1. Presentation Clarity: Data presentation is inconsistent, with some tables and figures lacking adequate labeling and explanation. For example, ensure all figures have descriptive captions explaining their relevance to the results.

Response: Thank you for your suggestion. We have revised both the Methods and Results sections accordingly. As stated in the Methods section, we extracted and presented data from each included article where available. Additionally, we have updated and completed the information provided in the tables and figures to ensure clarity and comprehensiveness. For the AI model with a more complex testing process, we included an additional table to capture and present all relevant information in a clear and organized manner. We believe these changes enhance the overall clarity and completeness of the manuscript.

  1. Comprehensiveness: Some key findings are discussed vaguely or omitted. A systematic comparison of the AI methods (e.g., accuracy, scalability, clinical applicability) should be included.

Response: Thank you for your feedback. Making a systematic comparison of the AI methods is challenging due to the varying stages of development of the tools, differences in evaluation approaches, and the diversity of input data and analytical methods used to generate predictions or results. Furthermore, this article is a narrative review, but we hope to have the possibility to make a systematic analysis. However, we appreciate your suggestion and have addressed this limitation in the Discussion section to clarify the constraints in directly comparing these AI methods.

Discussion:

  1. Key Findings and Interpretation: The discussion highlights AI’s potential but lacks depth in interpreting the key findings within the context of existing literature. Further analysis of why certain AI tools perform better in APL management is needed.

Response: Thank you for your suggestion. We have added a summary paragraph which states the benefit of AI tools in APL screening setting and the key benefits of its implementation.

  1. Limitations: While some limitations (e.g., small study sample) are briefly mentioned, others, such as biases in the reviewed studies and generalizability to different populations, are not adequately addressed.

Response: Thank you for your suggestion. We have enriched the discussion regarding the limitations of the proposed methods as instructed.

  1. Practical Implications: The broader implications of AI in improving diagnostic workflows and reducing healthcare disparities need further exploration. Propose concrete strategies for implementing these findings in clinical settings.

Response: Thank you for your suggestion. We have briefly added the most important key aspects of the perspectives of AI use in APL in the discussions section.

Conclusions:

  1. Addressing Aims: The conclusions are general and fail to explicitly address the original research aims. Clearly summarize how the study contributes to the understanding of AI applications in APL.

Response: Thank you for your suggestion. We have revised the conclusions to stress out the contribution of this paper to the current state of knowledge.

  1. Significance: The implications for future research and clinical practice are not well articulated. Provide actionable recommendations for researchers and clinicians to advance AI integration in oncology.

Response: Thank you for your suggestion. We have provided actionable recommendations as instructed.

The manuscript demonstrates significant linguistic issues that need thorough revision. Grammatical inconsistencies, including article misuse and incorrect verb tenses, impede readability. Sentence structures are often verbose, leading to redundancy and lack of clarity. Terminology, while technically accurate, is at times overly complex, making the content less accessible to a broader audience. Additionally, the transitions between sections are abrupt, affecting coherence. A comprehensive professional English editing is necessary to meet the standards of high-impact journals.

Once again, thank you very much for your work. We´ll be waiting for your answers about my comments.

Kindest regards,

Response: Thank you for your suggestion. The manuscript has been checked and polished by a native English speaker.

Comments on the Quality of English Language

The manuscript demonstrates significant linguistic issues that need thorough revision. Grammatical inconsistencies, including article misuse and incorrect verb tenses, impede readability. Sentence structures are often verbose, leading to redundancy and lack of clarity. Terminology, while technically accurate, is at times overly complex, making the content less accessible to a broader audience. Additionally, the transitions between sections are abrupt, affecting coherence. A comprehensive professional English editing is necessary to meet the standards of high-impact journals.

Response: Thank you for your suggestion. The manuscript has been checked and polished by a native English speaker.

We would like to thank you for your valuable comments which helped us improve the manuscript. All suggestions were taken into consideration and appropriate information, as well as required corrections, were provided. New/corrected parts are highlighted in yellow to facilitate the assessment of changes. We did our best to fulfil the expectations and we hope that you will be satisfied with our corrections. All in all, we thank you for your positive comments and appreciation regarding our manuscript.

Reviewer 2 Report

Comments and Suggestions for Authors

Dear authors,

I have now completed the review of the manuscript titled "Applications of Artificial Intelligence in Acute Promyelocytic Leukemia: an Avenue of Opportunities?"

In the present study,  the paper provides a valuable overview of emerging AI applications in APL diagnosis and management.

I have some suggestions to further improve the quality of the manuscript before its publication.

1. The paper relies heavily on retrospective analysis of previously published studies rather than conducting new primary research. While the authors identified 17 manuscripts for qualitative analysis, there appears to be no systematic assessment of study quality or risk of bias in the included papers. I would like authors to discuss articles like Consequential Advancements of Self-Supervised Learning (SSL) in Deep Learning Contexts, to overcome this shortcomings. This article would help understand the underlying machine learning techniques used in APL diagnostic tools, particularly as several systems mentioned in the review use self-supervised and weakly supervised approaches. Also, discussing articles like Recent Deep Learning-Based Brain Tumor Segmentation Models Using Multi-Modality Magnetic Resonance Imaging: A Prospective Survey would provide valuable background on deep learning applications in medical imaging analysis, which is highly relevant since the APL paper discusses several imaging-based AI diagnostic tools using blood smears and bone marrow samples.

2. Many of the AI tools described were developed and validated on relatively small datasets. For example, one study used only 51 APL cases for training, while another evaluated just 14 APL patients. These limited sample sizes raise questions about the generalizability and robustness of the proposed AI solutions. Also, the paper presents various performance metrics (AUROC, sensitivity, specificity) across different studies but does not adequately address the clinical significance of these numbers. Additionally, there is insufficient discussion of false positive and false negative rates, which are crucial considerations for diagnostic tools in acute leukemia.

Integration and Implementation Challenges

3. While the paper explores multiple AI applications, it provides limited discussion of the practical challenges of implementing these tools in clinical settings. Important considerations such as integration with existing laboratory workflows, cost-effectiveness, and required infrastructure are not thoroughly addressed. While focused on a different disease, discussing papers like Deep Learning Network Selection and Optimized Information Fusion for Enhanced COVID-19 Detection would provide insights into network architecture selection and data fusion approaches that could be applicable to APL detection systems. Also, OptEF-BCI: An Optimization-Based Hybrid EEG and fNIRS-Brain Computer Interface, though not directly related to APL, this paper discusses optimization of multi-modal data analysis systems, which could be relevant to understanding how to combine different diagnostic inputs (morphology, flow cytometry, etc.) for APL detection.

4. Most of the described AI tools lack robust external validation across different populations and healthcare settings. The paper acknowledges this limitation but does not fully explore its implications for real-world application. Finally, The research does not adequately demonstrate whether these AI tools actually improve patient outcomes or clinical decision-making compared to standard diagnostic approaches. There is limited discussion of comparative effectiveness with conventional methods.

Thank you for your valuable contributions to our field of research. I look forward to receiving the revised manuscript.

Author Response

Dear Editor in Chief,

Dear Academic Editor and Guest Editors,

We are very thankful to you and to the peer-reviewers for the pertinent notes; we have carefully read the comments and have revised/completed the manuscript accordingly. Our responses are given in a point-by-point manner below. All the changes to the manuscript are highlighted in yellow. We hope that, in this new form, the manuscript will be suitable for publication in the Journal of Clinical Medicine.

Dear authors, I have now completed the review of the manuscript titled "Applications of Artificial Intelligence in Acute Promyelocytic Leukemia: an Avenue of Opportunities?"

In the present study,  the paper provides a valuable overview of emerging AI applications in APL diagnosis and management.

I have some suggestions to further improve the quality of the manuscript before its publication.

  1. The paper relies heavily on retrospective analysis of previously published studies rather than conducting new primary research. While the authors identified 17 manuscripts for qualitative analysis, there appears to be no systematic assessment of study quality or risk of bias in the included papers. I would like authors to discuss articles like Consequential Advancements of Self-Supervised Learning (SSL) in Deep Learning Contexts, to overcome this shortcomings. This article would help understand the underlying machine learning techniques used in APL diagnostic tools, particularly as several systems mentioned in the review use self-supervised and weakly supervised approaches. Also, discussing articles like Recent Deep Learning-Based Brain Tumor Segmentation Models Using Multi-Modality Magnetic Resonance Imaging: A Prospective Survey would provide valuable background on deep learning applications in medical imaging analysis, which is highly relevant since the APL paper discusses several imaging-based AI diagnostic tools using blood smears and bone marrow samples.

 Response: Thank you for your suggestion. The study quality/risk of bias could not be ascertained because there are no specific instruments designed to evaluate these items accepted by Cochrane. Diagnosis of APL is not based on imaging techniques, however, a key role is played by cytomorphology as pointed out in our paper.

  1. Many of the AI tools described were developed and validated on relatively small datasets. For example, one study used only 51 APL cases for training, while another evaluated just 14 APL patients. These limited sample sizes raise questions about the generalizability and robustness of the proposed AI solutions. Also, the paper presents various performance metrics (AUROC, sensitivity, specificity) across different studies but does not adequately address the clinical significance of these numbers. Additionally, there is insufficient discussion of false positive and false negative rates, which are crucial considerations for diagnostic tools in acute leukemia.

Response: Thank you for your suggestion. We have sought to narrate the most significant details of the reviewed articles and checked for external validation, which was performed in more than half of the experiments described. All the methods displayed solid performances with low false positive and false negative rates as APL also harbors homogeneous changes in various parameters, fact that made this particular type of myeloid leukemia susceptible to algorithmic diagnosis via various ANN models. The clinical significance of these proposed models has been detailed in the introduction setting.

Integration and Implementation Challenges

  1. While the paper explores multiple AI applications, it provides limited discussion of the practical challenges of implementing these tools in clinical settings. Important considerations such as integration with existing laboratory workflows, cost-effectiveness, and required infrastructure are not thoroughly addressed. While focused on a different disease, discussing papers like Deep Learning Network Selection and Optimized Information Fusion for Enhanced COVID-19 Detection would provide insights into network architecture selection and data fusion approaches that could be applicable to APL detection systems. Also, OptEF-BCI: An Optimization-Based Hybrid EEG and fNIRS-Brain Computer Interface, though not directly related to APL, this paper discusses optimization of multi-modal data analysis systems, which could be relevant to understanding how to combine different diagnostic inputs (morphology, flow cytometry, etc.) for APL detection.

Response: Thank you for your suggestion. We have taken into consideration your suggestions in the revised version of the paper.

  1. Most of the described AI tools lack robust external validation across different populations and healthcare settings. The paper acknowledges this limitation but does not fully explore its implications for real-world application. Finally, The research does not adequately demonstrate whether these AI tools actually improve patient outcomes or clinical decision-making compared to standard diagnostic approaches. There is limited discussion of comparative effectiveness with conventional methods.

Response: Thank you for your suggestion. We have revised the Discussion section to more thoroughly address the lack of robust external validation for many of the described AI tools across diverse populations and healthcare settings. A paragraph describing the practical usage of the methods has also been added to the manuscript:

“Most of the methods described in the existing literature involve an ANN model implied in the process of APL detection. Experiments vary in terms of cohort sizes but so do the methods evaluated. A significant proportion of studies have been conducted on preexisting optical techniques which involve capturing blood films images and further data manipulation. The most explored experimental design uses a convolution neural network (CNN) which screens the optical image acquired. Even if the data sets from each reporting center is small, most of the studies report external validation with cross-sets provided by other institutions. A limitation worth mentioning is the high variation in data collection as pointed out by Barrera et al., given that staining and the image capture might vary across the commercially available staining and scanning laboratory devices. Lincz and his colleagues described how various iterations of AI tools achieved remarkable 98% rates of sensibility in identifying blast cells, at the expense of specificity which decreased to 12% in AI3, an acceptable trade-off for screening tools [LINCZ].

The graph neural networks (GNN) proved to be most suitable in more complex scenarios where multiparameter flow cytometry (MFC) has been explored as the source of data for the AI models. Cox et al. report in their paper which involved a small lot of patients (n=27 APL patients) promising 100% accuracy, a possible scenario as this method is not hindered by staining and acquisition factors [COX]. This result is encouraging for the unsupervised algorithms of MFC, though the real use-case in the screening setting is limited as immunophenotyping still involves sample preparation steps, making it less appealing in the screening setting. However, we cannot fail to recognize that such an AI component might benefit the laboratory in superior precision of the results, complementing a manual cytometry report. Nonetheless, another paper documents the possibility in reducing the analyzed parameters from 37 to just 3 (FSC-A, SSC-H, CD117), with similar AUROC performance [Monaghan]. While not yet a widespread quick identification method, MFC paired machine learning GNN structures might be integrated as aiding systems and also the correct marker selection might help reduce the cost of the procedure when processed through a previously trained ML.

Thank you for your valuable contributions to our field of research. I look forward to receiving the revised manuscript.

We would like to thank you for your valuable comments which helped us improve the manuscript. All suggestions were taken into consideration and appropriate information, as well as required corrections, were provided. New/corrected parts are highlighted in yellow to facilitate the assessment of changes. We did our best to fulfil the expectations and we hope that you will be satisfied with our corrections. All in all, we thank you for your positive comments and appreciation regarding our manuscript.

Round 2

Reviewer 1 Report

Comments and Suggestions for Authors

Dear Authors,

Thank you for submitting the revised version of your manuscript titled “Applications of Artificial Intelligence in Acute Promyelocytic Leukemia: An Avenue of Opportunities?”. I appreciate the effort made to address my previous comments. However, after a thorough review of the revised version, I find that several major concerns remain unaddressed, and further substantial revisions are required to meet the journal’s standards.

Introduction:

  1. Lack of clarity in research aims: While the research objectives have been slightly refined, they remain broadly stated. The study’s novelty and specific contributions beyond existing AI applications in leukemia are still not clearly articulated.
  2. Incomplete literature review: Some additional references have been included, but the background remains insufficient. Recent key studies on AI applications in hematology and oncology are missing, limiting the contextual strength of the manuscript. A more comprehensive review of the latest advancements is necessary.

Methods:

  1. Study design/protocol remains unclear: While the inclusion/exclusion criteria have been expanded, the methodology for selecting studies is still vague. Details on database sources, search strategies, and filtering processes need further elaboration.
  2. Lack of statistical methodology: No independent statistical analysis has been added. The manuscript briefly mentions performance metrics, but there is no justification for specific values (e.g., AUROC). Corrections for biases, validation strategies, or sensitivity analyses remain unexplored.
  3. Replicability concerns: The methods section still lacks sufficient detail to ensure reproducibility. The process by which AI models were assessed and validated is not fully described.

Results:

  1. Alignment with research aims is insufficient: Although the results are more structured, they still do not clearly demonstrate how AI improves APL diagnosis beyond traditional methods.
  2. Presentation clarity needs improvement: Some reorganization has been done, but redundancy between text, tables, and figures persists. Figure captions require further clarification.
  3. Lack of comprehensive comparisons: A systematic comparison of different AI models is still missing. The manuscript does not critically assess variations in model performance.

Discussion:

  1. Key findings lack in-depth interpretation: The discussion remains largely descriptive and does not critically assess the strengths and weaknesses of various AI methodologies.
  2. Insufficient discussion of limitations: While some limitations are mentioned, key concerns—such as potential biases, lack of external validation, and clinical implementation challenges—are still not fully addressed.
  3. Limited practical implications: The study does not provide clear recommendations for integrating AI into real-world clinical settings. A more detailed discussion on clinical applicability is needed.

Conclusions:

  1. Weak connection to research objectives: The conclusion summarizes the findings but does not explicitly address the study’s original aims.
  2. Lack of significance for future research: The broader impact of AI in APL management remains vague, and there are no concrete suggestions for future research directions or clinical translation.

References:

  1. Relevance and completeness: Although some additional references have been included, critical recent studies on AI in hematology remain missing. The reference list still contains outdated sources.

Language and Readability:

  1. Persistent grammatical and structural issues: The manuscript continues to exhibit grammatical errors, inconsistencies in verb tense, and awkward sentence structures.
  2. Terminology and clarity: Some technical terms remain overly complex or are used imprecisely, affecting readability and accessibility.

Although some minor revisions have been made, the manuscript still requires substantial improvements in methodology, statistical validation, and discussion depth before it meets the journal’s standards. I encourage you to make the necessary revisions to strengthen the scientific rigor and clarity of your work.

Once again, thank you for your efforts. We look forward to receiving your revised manuscript.

Kindest regards,

Comments on the Quality of English Language

Language and Readability:

Persistent grammatical and structural issues: The manuscript continues to exhibit grammatical errors, inconsistencies in verb tense, and awkward sentence structures.

Terminology and clarity: Some technical terms remain overly complex or are used imprecisely, affecting readability and accessibility.

Author Response

Dear Editor in Chief,

Dear Academic Editor and Guest Editors,

Dear Peer-Reviewer,

We are very thankful to you and for the pertinent notes; we have carefully read the comments and have revised/completed the manuscript accordingly. Our responses are given in a point-by-point manner below. All the changes to the manuscript are highlighted in yellow. We hope that, in this new form, the manuscript will be suitable for publication in the Journal of Clinical Medicine.

Thank you for submitting the revised version of your manuscript titled “Applications of Artificial Intelligence in Acute Promyelocytic Leukemia: An Avenue of Opportunities?”. I appreciate the effort made to address my previous comments. However, after a thorough review of the revised version, I find that several major concerns remain unaddressed, and further substantial revisions are required to meet the journal’s standards.

Response: Thank you for your suggestions. However, we must clearly point out that we were invited to submit a NARRATIVE review on the topic of AI in APL. The journal did not invite us to submit a systematic review or meta-analysis on this subject, and therefore we complied with what was asked from us.  

Introduction:

  1. Lack of clarity in research aims: While the research objectives have been slightly refined, they remain broadly stated. The study’s novelty and specific contributions beyond existing AI applications in leukemia are still not clearly articulated.

Response: Thank you for your suggestion. The aims have been further refined: “More specifically, our objective was to highlight the available integrated approaches that make appeal to conventional APL diagnostic instruments (cytomorphology, flow-cytometry, molecular biology/omics, and routine laboratory parameters) and AI, with a focus on AI branches such as ML, DL and CNNs and their performance metrics, in the assessment of APL. Therefore, we conducted a comprehensive narrative review of the current literature published between 2016 and 2025 and indexed in three reference scientific databases for medicine (PubMed/MEDLINE, Web of Science and SCOPUS) that investigates for the first time the applications of AI in APL.”      

  1. Incomplete literature review: Some additional references have been included, but the background remains insufficient. Recent key studies on AI applications in hematology and oncology are missing, limiting the contextual strength of the manuscript. A more comprehensive review of the latest advancements is necessary.

Response: Thank you for your suggestion. We understand the importance of providing adequate background; however, the purpose of our introduction was to set the necessary context for our specific focus on the use of AI in the diagnosis and management of APL. Our aim was not to provide an extensive review of AI applications across the entire field of hematology and oncology, but rather to center the discussion on AI’s role specifically in APL. Expanding the background to include a broader review of AI applications in oncology would shift the focus away from APL, which is the core of our manuscript, and place undue emphasis on AI algorithms in oncology as a whole. Additionally, our introduction does include relevant context on AI in hematology, with references predominantly from the past five years to ensure the inclusion of recent advancements. We believe that the current background provides sufficient contextual strength for our specific focus on AI applications in APL and aligns with the scope of our manuscript.  In response to this comment, we have also added further references to strengthen the background and ensure a more comprehensive overview related to our specific focus on AI applications in APL.

New paragraph added reads as follows: “Recent years have brought undeniable progress in the diagnosis and management of hematological malignancies. Moreover, the application of artificial intelligence (AI) and its branches, such as machine learning (ML) or deep learning (DL), in the field of medicine, including hematology, has brought to light new avenues for research in the fields of leukemia, lymphoma, myeloma, myeloproliferative (MPN) or myelodysplastic neoplasms (MDS), to name a few [11]. There seems to be a growing interest towards AI in the evaluation of hematologic malignancies as depicted in Figure 1.

Figure 1. Publication trends on the use of AI in malignant hematology. The number of manuscripts indexed in PubMed/MEDLINE [12] and related to this topic has steadily increased since the year 2017. 

 Examples of AI applications in hematology include but are not limited to the use of systems designed to analyze images in cytomorphology and pathology, of bioinformatics in order to refine the results of genomics and other omics-based approaches, of language processing systems, utilized to rapidly assess real-world data such as electronic health records, of decision support systems meant to propose diagnostic and treatment algorithms, as well as of medical-device related systems, e.g., smartwatches with sensors capable of detecting changes in body parameters and signal potential treatment-related complications [13]. ML has proven excellent performance metrics (sensitivity, specificity, and accuracy) in the diagnosis, classification and prognostication of MDS and AML using a diverse range of information obtained from clinical examinations as well as laboratory data, i.e., cytomorphology (peripheral blood and bone marrow aspirate smears), cytogenetics, RNA sequencing and/or genomics [13, 14]. The assessment of peripheral and blood marrow smear images using AI, and in particular convolutional neural networks (CNNs), demonstrates over 90% accuracy in diagnosing AML and MDS, as well as distinguishing between these two myeloid malignancies and chronic myeloid leukemia (CML), chronic lymphocytic leukemia (CLL) or acute lymphoblastic leukemia (ALL) [13]. Moreover, AI has proven useful in estimating prognosis in MDS and AML. Researchers have made appeal to gradient-boosting to establish decision tree machine learning libraries that could predict MDS development one year before the actual diagnosis, transformation of MDS to AML, overall survival or complete remission, or inform about AML prognosis solely based on FLT3-ITD gene mutation status [14]. In addition, AI can be used to predict treatment response, with CNNs, decision trees, operators and classifiers being employed to predict response to hypomethylating agents, allogeneic bone marrow transplantation, or the risk of AML relapse [14]. A comprehensive review on the applications of AI in MDS highlighted that this innovative technique can be used to assess complete blood counts, peripheral blood and bone marrow aspirate smears, as well as flow-cytometry data, and establish a diagnosis of MDS, differentiate MDS from aplastic anemia or AML, and detect dysplastic cells of all lineages as well as myeloblasts. This is of particular importance as the differential diagnosis between hypoplastic MDS and aplastic anemia is often cumbersome and both diseases require different treatment strategies. Additionally, establishing myeloblast thresholds is extremely relevant in distinguishing MDS from AML and in selecting the most appropriate therapy to prevent leukemic transformation [14]. CNNs were the most frequent AI models used in the evaluation of MDS, were internally validated and displayed excellent performance metrics and lower percentages of misclassifications. However, limitations of AI in MDS included lack of external validation and of standardized methodology for the use of flow-cytometry in MDS, small sample sizes, as well as the need for supervision [14]. Furthermore, AI has proven significant potential in AML management, as it is able to discriminate between hematopoietic stem cells and leukemia stem cells based on cytomorphology and flow-cytometry reports with an accuracy of nearly 93.5%. These findings further solidify the hypothesis that AI can become practical in the prediction of response to AML treatment, including early relapse post-allogeneic hematopoietic stem cell transplantation [15]. Complications can also be predicted using AI algorithms in AML. Doknic and collaborators developed a ML-based method to assess the risk of venous thrombosis in AML, revealing that male sex, history of thrombotic events,   international normalized ratio, hematopoietic cell transplantation-specific comorbidity index and intensive chemotherapy are risk factors for the onset of thrombotic complications in this myeloid malignancy [16]. In children, AI has been employed to quantify the risk of viral and bacterial infections in AML and ALL with an accuracy of nearly 80%. Using a framework based on ML (decision trees combined with SemNet 2.0), Al-Hussaini and his team have demonstrated that there are several factors which can influence the risk of infectious complications in pediatric patients with ALL or AML: race, type of acute leukemia, association with Down syndrome or lupus, central nervous system involvement at diagnosis, National Cancer Institute risk group, chemotherapy regimen and course, glucose, zinc or iron metabolism, growth factor levels [17].

Similarly, AI seems promising in the evaluation of MPN as well. In BCR::ABL1-negative MPN, ML and CNN have displayed extremely good performance metrics in diagnosing MPN, differentiating between MPN subtypes, and in particular distinguishing essential thrombocythemia from prefibrotic primary myelofibrosis, as well as in predicting prognosis, occurrence of thrombotic events or treatment response to hydroxyurea. These AI tools made appeal to clinical data, routine laboratory parameters, peripheral blood smears, bone marrow biopsy samples, transcriptomics and genomics, and led to rapid assessment of cases and accurate discrimination between MPN subtypes. However, limitations to the AI instruments in MPN include the need for larger patient cohorts for training and validation, as well as the requirement for human supervision for many of these tools [18]. More recent investigations have delineated that AI can accurately predict the development of thrombosis in subjects with polycythemia vera who were prescribed treatment with hydroxyurea. In the aforementioned AI algorithm, the onset of thrombotic complications could be predicted by using only three complete blood count-derived variables: red cell distribution width, neutrophil percentage and lymphocyte percentage [19]. In CML, ML instruments have been applied to investigate the interactions established between the BCR-ABL1 protein and tyrosine kinase inhibitors in the scope of discovering novel drugs and therapeutic targets [20].     

Furthermore, the use of AI platforms has also been warranted in the assessment of chronic lymphoproliferative disorders. In multiple myeloma (MM), AI has been employed to establish an early diagnosis, prognosis, disease staging, identification of bone lesions and molecular aberrations, as well as to guide therapy selection [21]. For example, CNNs and decisional trees have been combined with data obtained from routine hematological and biochemical blood tests, flow-cytometry and spectroscopy to accurately establish an early diagnosis in MM, as well as to stage the disease [21]. Moreover, information collected from imaging (PET-CT, CT or MRI) has been connected with AI techniques (CNNs, random forest, support vector machines) to quantify the level of skeletal involvement in MM, as well as to discriminate between bone lesions and vertebral metastases of other causes [21]. Response to treatment has also been estimated with the aid of AI. Random forest, support vector machines, simulations of treatment learning signatures and multilearning treatment approaches have been combined with clinical and genomic data to estimate response to proteasome inhibitors, immunomodulatory agents, corticosteroids and chemotherapy [21]. ML models have also been developed to predict the risk of infectious complications which account for a significant proportion of deaths in MM. Aided by the use of random forest, decision tree and gradient boosting algorithms, Mikulski and team discovered that low platelet distribution width values predict the occurrence of pneumonia in patients with MM who had received regimens containing the anti-CD38 monoclonal antibody daratumumab [22]. Concerning lymphoma, a recent meta-analysis of diagnostic studies has highlighted that AI algorithms displayed a specificity and sensitivity of 94% and 87%, respectively, in the detection of lymphoma based on imaging data [23]. PET-CT findings have also been combined with ML models to decide whether there is a degree of bone marrow involvement in Non-Hodgkin’s lymphoma, a piece of information which is crucial for disease staging and selection of the most appropriate therapy [24]. Moreover, proteomics has been connected with bioinformatics to quantify the risk of progression of follicular lymphoma. Using unsupervised ML models, Hemmingsen et al have revealed that patients with low expression of STING1 and IDH2 are at higher odds of progressing to diffuse large B-cell lymphoma (DLBCL) [25]. In addition, AI seems to be superior to the reports given by expert pathologists in predicting transformation of CLL or follicular lymphoma to DLBCL [26]. ML and natural language processing have also been employed to analyze the electronic health records of over 500 CLL patients from Spain in order to produce real-world evidence of the clinical characteristics and therapy patterns from this country in an attempt to solidify the argument that there is a need for personalized treatment in hematological malignancies [27]. Another field of hematology where AI has been investigated is hematopoietic stem cell transplantation (HSCT). Random forest and decision tree-based approaches have successfully appreciated the risk of relapse and the prognosis of patients who had undergone this procedure and it has been hypothesized that ML will be able in the near future to assist hematologists in selecting the optimal conditioning regimen in HSCT [28]. For example, the relapse rate at 2 years following allogeneic HSCT for AML or ALL in children has been estimated using ML. The random forest approach exhibited an accuracy of 85% for ALL and 81% for AML, a sensitivity of 85% for ALL and 75% for AML, and a specificity of 89% for ALL and 100% for AML [29]. AI methods have also enhanced the prediction of post-HSCT complications, e.g., the onset of acute kidney injury. Musial and collaborators applied a random forest classifier to a dataset of pediatric subjects who had undergone HSCT and concluded that the estimated glomerular filtration rate pre-HSCT and in the early period following HSCT, administration of methotrexate use, acute graft versus host disease and the development of viral infections are the most accurate predictors of acute kidney injury in HSCT recipients [30].

Methods:

  1. Study design/protocol remains unclear: While the inclusion/exclusion criteria have been expanded, the methodology for selecting studies is still vague. Details on database sources, search strategies, and filtering processes need further elaboration.

Response: Thank you for your suggestion. The Methods section has been rewritten as follows: The current review was conducted in agreement with the “Preferred Reporting Items for a Systematic Review and Meta-Analysis” (PRISMA) norms [35]. The research question behind the elaboration of this manuscript was: “Can AI be useful in the assessment of APL?”. 

Information sources and search strategy. We developed a search strategy in three databases of reference for medicine (PubMed/MEDLINE, SCOPUS and Web of Science) using the following specific keywords and word combinations: (“acute promyelocytic leukemia” OR “acute promyelocytic leukaemia”) AND (“artificial intelligence” OR “machine learning” OR “deep learning” OR “neural networks” OR “clinical decision support system” OR “computer vision” OR “digital health” OR “large language models” OR “random forest”). The Polygot Search Translator was used to convert the search strategy query between databases [36]. No restrictions were applied to the search strategy in terms of language. In terms of publication timeline, we included manuscripts published from the inception of the aforementioned databases until February 10th, 2025.

Eligibility criteria. The following inclusion criteria were applied: 1. the included manuscripts were either original research articles or research letters; 2. the study groups of these investigations included confirmed APL cases; 3. the examined studies investigated the use of any AI branch (ML, DL, CNNs, etc.) in APL; 4. the manuscripts were published in a language known to the authors of the current manuscript (English, French, Italian, German, Romanian); 5. the full-texts of the papers could be retrieved. We opted for the following exclusion criteria: 1. the manuscripts were excluded if they were designed as reviews, book chapters, case reports, meeting abstracts, animal studies or other types of manuscripts; 2. studies conducted on laboratory animals; 3. the diagnosis of APL was not confirmed, e.g., the subjects were diagnosed with non-APL AML or APL-like AML; 4. the papers were published in a language not spoken by the authors; 5. the full-texts of the manuscripts could not be retrieved; 6. the investigations did not report sufficient data on the outcomes of interest (AI models, performance metrics). 

Study selection, data collection, outcomes of interest, risk of bias, effect measures and synthesis methods. Three researchers independently assessed the titles and abstracts of the selected publications according to the aforementioned search strategy. Quality assessment was conducted using the Quality Assessment of Diagnostic Accuracy Studies version 2 (QUADAS-2) instrument [37]. Data was centralized and analyzed using Microsoft Office Excel in dedicated spreadsheets. The following information was extracted from the original studies: the type of AI model developed, the input data required to generate predictions/results, the training, testing, and/or validation datasets (data description and number of samples), the reported characteristics of performance for the AI model - in relation to the stage of development and/or testing. The main performance metrics considered were: sensitivity (the rate of true positive responses), specificity (the rate of true negative answers), accuracy (the sum of true positive and true negative responses divided by the number of total observations), and the AUROC (area under the receiver operating characteristic curve). When available, other performance characteristics, e.g., precision, were collected. We used the performance metrics to briefly compare the performance of the AI model clusters investigated by the authors, outlining the subtle differences. If unavailable and data allowed it, we calculated the performance metrics based on the following formulas [38]:

  • Sensitivity (True Positive Rate, TPR) = (True Positives)/(True Positives + False Negatives) (it measures the proportion of actual positives that are correctly identified)
  • Specificity (True Negative Rate, TNR) = (True Negatives)/(True Negatives + False Positives) (the ratio of correctly predicted negative observations to all actual negatives; it measures the proportion of actual negatives that are correctly identified by the model)
  • Accuracy = (True Positives + True Negatives)/Total Observations (the ratio of correctly predicted observations to the total observations; a common measure of the overall performance of a classification model)
  • AUROC = a measure of the ability of a classifier to distinguish between classes. It plots the TPR against the False Positive Rate (FPR) at various threshold settings. It is positively correlated with the performance of the analyzed model.

  1. Lack of statistical methodology: No independent statistical analysis has been added. The manuscript briefly mentions performance metrics, but there is no justification for specific values (e.g., AUROC). Corrections for biases, validation strategies, or sensitivity analyses remain unexplored.

Response: Thank you for your suggestion. We conducted a quantitative assessment of performance metrics. See Results section, subsections 3.6, 3.7, 3.8 and 3.9 and Tables 3, 4, 5, 6, 7.

  1. Replicability concerns: The methods section still lacks sufficient detail to ensure reproducibility. The process by which AI models were assessed and validated is not fully described.

 Response: Thank you for your suggestion. The Methods section has been rewritten.

Results:

  1. Alignment with research aims is insufficient: Although the results are more structured, they still do not clearly demonstrate how AI improves APL diagnosis beyond traditional methods.

Response: Thank you for your suggestion. As stated in the scope of this article, the AI focus in this particular type of leukaemia is centred on the rapid diagnostic potential. In comparison with other types of AML, APL requires immediate treatment with differentiation agents, namely ATRA (all-trans retinoic acid) and also has unique distinctive features (either morphological elements on the blood smears or particular changes in biochemistry and coagulation parameters). The real use case of AI in APL resides in novel screening techniques which might prompt a specific case flagging, followed by further confirmatory review by a pathologist. The narrative review tried to emphasise from each study the method proposed for this rapid and efficient screening, listing also briefly the type of ANN implied (CNN, GNN etc.). APL is a particular type of AML and is a good candidate for automation as it has certain features that enable its differentiation from the other acute myeloid leukemias. 

  1. Presentation clarity needs improvement: Some reorganization has been done, but redundancy between text, tables, and figures persists. Figure captions require further clarification.

Response: Thank you for your suggestion. As this is a review, the results are inherently presented as a detailed text to provide a comprehensive synthesis of the existing literature. To facilitate readability, we have included tables and figures to summarize key points. However, due to the extensive context and large amount of information covered in our review, we believe that presenting specific data solely in tables or figures would disrupt the narrative flow and potentially reduce the reader’s understanding of the context. 

Given the breadth of information, we aimed to balance textual explanation with visual summaries, but omitting details from the text entirely in favor of tables alone would limit the context needed to interpret the data accurately. We have made efforts to minimize redundancy where possible while ensuring that both the text and supplementary materials remain informative and coherent.

  1. Lack of comprehensive comparisons: A systematic comparison of different AI models is still missing. The manuscript does not critically assess variations in model performance.

Response: Thank you for your suggestion. We have quantitatively assessed the performance metrics of the AI models used for APL assessment as instructed.  However, a meta-analysis was not performed because the data were heterogeneous. A meta-analysis requires a degree of homogeneous data—studies with consistent methodologies, similar outcomes, and comparable models—so that statistical comparisons can be made. In our case, the wide variety of AI tools, development stages, purposes, and evaluation methods naturally lends itself to a review rather than a meta-analysis. Our work provides an overview of a heterogeneous body of research, which is exactly what a review is meant to do.

Discussion:

  1. Key findings lack in-depth interpretation: The discussion remains largely descriptive and does not critically assess the strengths and weaknesses of various AI methodologies.

Response: Thank you for your suggestion. This section has been improved. However, we must point out that the quantitative assessment of performance metrics reveals that the proposed AI models can efficiently be used in APL assessment as hypothesized.

The road to the implementation of AI tools in APL diagnosis in healthcare facilities is long and challenging. While the majority of the studies reported excellent performance characteristics, most of these tools are in the early stages of development and testing. With few exceptions, the majority of the AI tools with potential applications in APL management were tested on small datasets, lacked external validation, and were evaluated mostly outside of the clinical settings. While some models were trained and tested with many thousands of images, these images were obtained from a relatively small group of APL patients. With APL diagnosis being an emergency and implying immediate therapeutic and supportive interventions, the safety of implementing AI tools in clinical practice with this purpose must be thoroughly evaluated. The above-described studies reported promising results, but further research is required to bring technological progress closer to the hematology clinics. Another limitation is that flow-cytometry and omics-based techniques to diagnose APL are not widely available worldwide, with low-income economies encountering potential barriers in implementing AI-guided APL screening. Moreover, in order to train AI software, there is a need for adequate preanalytical processing of biological samples as well as experts in APL and bioinformaticians. It is also noteworthy to mention that several ML systems used in APL diagnostics use self-supervised and weakly supervised approaches that are prone to errors as previously demonstrated elsewhere [60].

From a statistical perspective, the types of input data and the subsequent ANN models used proved to have their own strengths, worthy to be integrated as complementary tools in the diagnostic process of APL. Peripheral blood smears-based AI assessment seemed to offer a balanced approach, while bone marrow aspirate smears-guided AI instruments resulted in higher sensitivity rates. ANN-conducted interpretations of other laboratory data ensured elevated accuracy and specificity. Further research is needed to verify whether combinations of these AI methods might improve the accuracy of APL diagnosis prior to a standard hematopathology evaluation.  

The current narrative review indicates that efforts have been made to develop several AI models with applications in the diagnosis and case management of APL. While most of them might seem to perform similarly, a comparison is challenging in the context of the varying stages of development of the tools, differences in evaluation approaches, and the diversity of input data and analytical methods used to generate predictions or results. In conclusion, the results of our analysis suggest that AI can emerge as a relevant tool in the evaluation of APL cases and potentially contribute to a more rapid screening and identification of this hematological emergency. AI models based on cytomorphology displayed the highest specificity, accuracy and AUROC for peripheral blood smears, and the highest sensitivity for bone marrow aspirate smears. AI methods based on other biomarkers exhibited superior specificity and accuracy but lower sensitivity and AUROC values. 

  1. Insufficient discussion of limitations: While some limitations are mentioned, key concerns—such as potential biases, lack of external validation, and clinical implementation challenges—are still not fully addressed.

Response: Thank you for your suggestion. This section has been improved. However, we must point out that the quantitative assessment of performance metrics reveals that the proposed AI models can efficiently be used in APL assessment as hypothesized.

The road to the implementation of AI tools in APL diagnosis in healthcare facilities is long and challenging. While the majority of the studies reported excellent performance characteristics, most of these tools are in the early stages of development and testing. With few exceptions, the majority of the AI tools with potential applications in APL management were tested on small datasets, lacked external validation, and were evaluated mostly outside of the clinical settings. While some models were trained and tested with many thousands of images, these images were obtained from a relatively small group of APL patients. With APL diagnosis being an emergency and implying immediate therapeutic and supportive interventions, the safety of implementing AI tools in clinical practice with this purpose must be thoroughly evaluated. The above-described studies reported promising results, but further research is required to bring technological progress closer to the hematology clinics. Another limitation is that flow-cytometry and omics-based techniques to diagnose APL are not widely available worldwide, with low-income economies encountering potential barriers in implementing AI-guided APL screening. Moreover, in order to train AI software, there is a need for adequate preanalytical processing of biological samples as well as experts in APL and bioinformaticians. It is also noteworthy to mention that several ML systems used in APL diagnostics use self-supervised and weakly supervised approaches that are prone to errors as previously demonstrated elsewhere [60].

From a statistical perspective, the types of input data and the subsequent ANN models used proved to have their own strengths, worthy to be integrated as complementary tools in the diagnostic process of APL. Peripheral blood smears-based AI assessment seemed to offer a balanced approach, while bone marrow aspirate smears-guided AI instruments resulted in higher sensitivity rates. ANN-conducted interpretations of other laboratory data ensured elevated accuracy and specificity. Further research is needed to verify whether combinations of these AI methods might improve the accuracy of APL diagnosis prior to a standard hematopathology evaluation.  

The current narrative review indicates that efforts have been made to develop several AI models with applications in the diagnosis and case management of APL. While most of them might seem to perform similarly, a comparison is challenging in the context of the varying stages of development of the tools, differences in evaluation approaches, and the diversity of input data and analytical methods used to generate predictions or results. In conclusion, the results of our analysis suggest that AI can emerge as a relevant tool in the evaluation of APL cases and potentially contribute to a more rapid screening and identification of this hematological emergency. AI models based on cytomorphology displayed the highest specificity, accuracy and AUROC for peripheral blood smears, and the highest sensitivity for bone marrow aspirate smears. AI methods based on other biomarkers exhibited superior specificity and accuracy but lower sensitivity and AUROC values.

  1. Limited practical implications: The study does not provide clear recommendations for integrating AI into real-world clinical settings. A more detailed discussion on clinical applicability is needed.

Response: Thank you for your suggestion. This section has been enriched with other suggestions as instructed: Future research should focus on the selection of the best AI method to diagnose APL, reduce the rate of false negatives and false positives, as well as investigate the potential benefits of multi-modal data analysis systems which could be relevant to understanding how to combine different diagnostic inputs, e.g., complete blood count data, morphology, flow-cytometry, omics, and others. Moreover, international collaboration is needed to create a large, freely and widely available APL dataset that could serve to validate AI methods. Future investigations should explore the use of other CBC-derived indices, biochemical marker, hemostasis screening tests and cytomorphology in the diagnosis of APL, as well as in the prediction of its complications, for example to predict the risk of disseminated intravascular coagulation, infections or treatment-related side effects. This would be of particular importance in resource-limited settings where MFC, FISH or molecular biology are not widely available.  

Conclusions:

  1. Weak connection to research objectives: The conclusion summarizes the findings but does not explicitly address the study’s original aims.

Response: Thank you for your suggestion. This section was revised as instructed: In conclusion, the results of our analysis suggest that AI can emerge as a relevant tool in the evaluation of APL cases and potentially contribute to a more rapid screening and identification of this hematological emergency. AI models based on cytomorphology displayed the highest specificity, accuracy and AUROC for peripheral blood smears, and the highest sensitivity for bone marrow aspirate smears. AI methods based on other biomarkers exhibited superior specificity and accuracy but lower sensitivity and AUROC values.

  1. Lack of significance for future research: The broader impact of AI in APL management remains vague, and there are no concrete suggestions for future research directions or clinical translation.

Response: Thank you for your suggestion. This section has been enriched with other suggestions as instructed: Future research should focus on the selection of the best AI method to diagnose APL, reduce the rate of false negatives and false positives, as well as investigate the potential benefits of multi-modal data analysis systems which could be relevant to understanding how to combine different diagnostic inputs, e.g., complete blood count data, morphology, flow-cytometry, omics, and others. Moreover, international collaboration is needed to create a large, freely and widely available APL dataset that could serve to validate AI methods. Future investigations should explore the use of other CBC-derived indices, biochemical marker, hemostasis screening tests and cytomorphology in the diagnosis of APL, as well as in the prediction of its complications, for example to predict the risk of disseminated intravascular coagulation, infections or treatment-related side effects. This would be of particular importance in resource-limited settings where MFC, FISH or molecular biology are not widely available.  

References:

  1. Relevance and completeness: Although some additional references have been included, critical recent studies on AI in hematology remain missing. The reference list still contains outdated sources.

Response: Thank you for your suggestion. We have discussed in more detail other recent studies on AI in hematology. Please see the introduction. We would like to point out that all studies on AI in hematology have been published in the last 2-3 years, thus they are clearly not outdated. If older manuscripts were included, they were key reference papers for the hematology field.

Language and Readability:

  1. Persistent grammatical and structural issues: The manuscript continues to exhibit grammatical errors, inconsistencies in verb tense, and awkward sentence structures.

Response: Thank you for your suggestion. We have carefully reviewed the manuscript using advanced grammar tools and addressed minor issues prior to submission. Additionally, another reviewer has confirmed that the language is clear and does not require further modifications. Nonetheless, we have conducted another thorough review of the text to ensure clarity and readability. We believe the current version reflects a high standard of English suitable for publication.

  1. Terminology and clarity: Some technical terms remain overly complex or are used imprecisely, affecting readability and accessibility.

Response: Thank you for your suggestion. We have carefully reviewed the manuscript using advanced grammar tools and addressed minor issues prior to submission. Additionally, another reviewer has confirmed that the language is clear and does not require further modifications. Nonetheless, we have conducted another thorough review of the text to ensure clarity and readability. We believe the current version reflects a high standard of English suitable for publication.

Although some minor revisions have been made, the manuscript still requires substantial improvements in methodology, statistical validation, and discussion depth before it meets the journal’s standards. I encourage you to make the necessary revisions to strengthen the scientific rigor and clarity of your work.

Once again, thank you for your efforts. We look forward to receiving your revised manuscript.

Response: Thank you for your suggestions. We hope that the revised manuscript is suitable for publication in the Journal of Clinical Medicine.

Comments on the Quality of English Language - Language and Readability: Persistent grammatical and structural issues: The manuscript continues to exhibit grammatical errors, inconsistencies in verb tense, and awkward sentence structures. Terminology and clarity: Some technical terms remain overly complex or are used imprecisely, affecting readability and accessibility.

Response: Thank you for your suggestion. The manuscript has been checked by a native English speaker in addition to Grammarly. If minor errors remain, they can be fixed during copy-editing and proofreading if the paper is accepted. 

Reviewer 2 Report

Comments and Suggestions for Authors

All comments were addressed.

Author Response

Thank you for your constructive feedback and for considering that our manuscript is now suitable for publication.